# Detection of the cloud liquid water path horizontal inhomogeneity in a coastline area by means of ground-based microwave observations: feasibility study

Vladimir S. Kostsov [1], Dmitry V. Ionov [1], and Anke Kniffka [2]

[1] Department of Atmospheric Physics, Faculty of Physics, St. Petersburg State University, Russia

[2] Zentrum für Medizin-Meteorologische Forschung, Deutscher Wetterdienst, Freiburg, Germany

*Correspondence to: Vladimir S. Kostsov (v.kostsov@spbu.ru)*

**Abstract.** The improvement of cloud modelling for global and regional climate and weather studies requires comprehensive information on many cloud parameters. This information is delivered by remote observations of clouds from ground-based and space-borne platforms using different methods and processing algorithms. Cloud liquid water path (LWP) is one of the main obtained quantities. Previously, the measurements of LWP by the SEVIRI and AVHRR satellite instruments provided the evidences of the systematic differences between LWP values over land and water areas in Northern Europe. An attempt is made to detect such differences by means of ground-based microwave observations performed near the coastline of the Gulf of Finland in the vicinity of St.Petersburg, Russia. The microwave radiometer RPG-HATPRO located 2.5 km from the coastline is functioning in the angular scanning mode and is probing the air portions over land (at elevation angle 90°) and over water area (at 7 elevation angles in the range 4.8°-30°). The problem of the LWP horizontal gradient detection is examined in the measurement domain: the brightness temperatures of the microwave radiation measured at different elevation angles in the 31.4 GHz and 22.24 GHz spectral channels are analysed and compared with the corresponding values which were calculated under the assumption of horizontal homogeneity of the atmosphere. Several specific cases, selected on the basis of the analysis of the satellite observations by the SEVIRI instrument were considered in detail including: clear-sky conditions, the presence of clouds over the radiometer and at the same time the absence of clouds over the Gulf of Finland, and overcast conditions over the radiometer and over the opposite shore of the Gulf of Finland. The influence of the land-sea LWP difference on the brightness temperature values in the 31.4 GHz spectral channel has been demonstrated and the following features have been detected: (1) an interfering systematic signal is present in the 31.4 GHz channel which can attributed to the humidity horizontal gradient; (2) clouds over the opposite shore of the Gulf of Finland mask the LWP gradient effect. Preliminary results of the retrieval of LWP over water by statistical regression method applied to the microwave measurements by HATPRO in the 31.4 GHz and 22.24 GHz channels are presented. The monthly averaged results are compared to the corresponding values derived from the satellite observations by the SEVIRI instrument and from the reanalysis data. The SEVIRI and the HATPRO instruments detect positive LWP land-sea gradients during all seasons but the magnitude of the gradient detected by the ground-based instrument is considerably smaller than detected by the satellite

instrument. The LWP gradients provided by HATPRO and reanalysis during warm season are in a very good agreement. During cold season in contrast to the SEVIRI and the HATPRO data, the reanalysis data demonstrate negative LWP gradient.

**Keywords:** cloud liquid water path; remote sensing; ground-based microwave radiometer; RPG-HATPRO; horizontal gradients of atmospheric parameters

## 1 Introduction

The improvement of global/regional climate/weather forecasting models requires comprehensive information on atmospheric composition, physical and chemical processes, and in particular the information on interactions between different components of the climate system: the atmosphere, water areas, land surfaces, snow and ice cover, and biosphere. Boe and Terray (2014) analysed the role of soil-atmosphere interactions, cloud-temperature interactions and land-sea warming contrast in summer European climate change. High resolution regional climate models were used (25 km) with a good realism of orography and coasts that could help in reducing the biases in local climate existing in low-resolution GCM simulations. The study by Fersch et al. (2019) has been devoted to the exchange of water, trace gases and energy between land surface and atmospheric boundary layer. This study examined the ability of the hydrologically enhanced version of the Weather Research and Forecasting Model (WRF-Hydro) to reproduce the regional water cycle by means of a two-way coupled approach and assessed the impact of hydrological coupling with respect to a traditional regional atmospheric model setting. One of the important parts of the climate system is cloud cover. Its variations significantly (and immediately) alter the heat balance of the earth's climate system on an hourly time scale, but their effects are profound from seasonal through decadal timescales, therefore the physical processes involving cloudiness–water vapor–surface temperature interaction need further investigation (Groisman et al., 2000). Tang et al. (2012) have shown that the variance of European summer temperature is partly explained by changes in summer cloudiness. Europe has become less cloudy (except northeastern Europe) and the regions east of Europe have become cloudier in summer daytime. However, the results obtained by Tang et al. (2012) suggest that the cloud cover is either the important local factor influencing the summer temperature changes in Europe or a major indicator of these changes.

Clouds, as an important climate influencing factor, are described by a large number of parameters of micro and macro-physics. Cloud liquid water path (LWP) is one of the main quantities being a measure of the total mass of the liquid water droplets in the atmosphere above a unit surface area on the earth, given in units of kg m$^{-2}$. The information on LWP is delivered mainly by remote observations of clouds from ground-based and space-borne platforms using different methods and processing algorithms. The principal space-borne techniques are based on the derivation of LWP from measurements of atmospheric self-emitted microwave (MW) radiation or from measurements of the reflected sunlight in visible and near-infrared ranges. The MW satellite sensors perform LWP measurements during day and night but only over water areas since

the emissivity of the land surface is highly variable. The advantage of the satellite instruments which register the reflected solar radiation in visible and near-infrared ranges is the ability to make observations over water areas and land surface as well (however only in the day time). Two instruments of this type are well-known: SEVIRI (Spinning Enhanced Visible and InfraRed Imager) and AVHRR (Advanced Very High Resolution Radiometer). The description of the information products delivered by these instruments and relevant to cloud properties can be found in the papers by Stengel et al. (2014, 2017).

Previously, the measurements of LWP by the satellite instruments SEVIRI and AVHRR provided the evidences of the differences between LWP values over land and water areas in Northern Europe. The data from the AVHRR instrument were used for compiling regional cloud climatology for the Scandinavian region (Karlsson, 2003). Analysis of this climatology has shown that during spring and summer the cloud amount over land in this region is larger than the cloud amount over the Baltic Sea and major lakes. Karlsson (2003) explained this phenomenon by the stabilization of near-surface layer of the troposphere over water bodies due to air cooling by the cold fresh water from melting snow. This explanation is in a good agreement with the fact revealed later in the study by Kostsov et al. (2018b): the land-sea gradient of the mean LWP values detected by the SEVIRI instrument in the vicinity of St.Petersburg (Russia) for the cold season was noticeably lower than for the warm season. St.Petersburg is located at the estuary of the Neva River which flows in the Gulf of Finland. The magnitude of the land-sea difference for mean LWP values obtained by SEVIRI in this area for the two-year period of 2013-2014 was about 0.040 kg m$^{-2}$, which was about 50 % relative to the mean value over land.

In general, the investigation of cloud properties in the coastal zones is an interesting and important task due to presence of specific atmospheric processes, for example sea breezes, which are able to generate clouds. The climatological study of the impact of sea breezes on cloud types was done by Azorin-Molina et al. (2008) for the area in the southeast of the Iberian Peninsula (province of Alicante, Spain) and for the 6-year period (2000-2005) based on cloud observations at synoptic station. The authors of mentioned study emphasize that their findings are site-specific and should be similar to other coastal locations, however, cloud formation associated with sea breezes is also influenced by geographical-physical, meteorological, hydrological and oceanic factors. Therefore there is a need for further research. The sea breeze effects were studied also on the basis of data derived from space-borne observations by AVHRR instrument (Azorin-Molina et al., 2009).

The satellite instruments working in visible and near-infrared ranges are very sensitive to the observational conditions. There are specific requirements to SEVIRI observations: measurements are restricted just after sunrise and before sunset when the solar zenith angle (SZA) is too large. Therefore, all SEVIRI measurements when SZA was greater than 72° were excluded from consideration in the studies by Roebeling et al. (2008) and Kostsov et al. (2018b). As a result, in the latter study devoted to the LWP measurements at high latitudes (60°N) no measurements during winter months December and January could be selected for analysis, and the number of measurements selected in February and suitable for analysis was very small. Besides, the problem of the misinterpretation of measurements in winter over the snow-covered and ice-covered surfaces with high reflectance should be mentioned (Musial, 2014; Kostsov et al., 2019). So, the considered satellite

observations are impossible in the night time, in winter at Northern latitudes, and there may be problems in winter in the day time over the snow- and ice-covered surfaces. Therefore, in the present study an attempt was made to find a kind of a supplement to satellite measurements in a coastal area in the form of detection of the land-sea LWP gradients by means of ground-based microwave observations. The concept of these measurements is straightforward: a radiometer which is located close to a coastline can probe the air portions over land and water surface if it works in the angular scanning mode at appropriate direction. Microwave measurements can be carried out during all seasons, day and night, excluding rain and strong snowfall conditions. Ground-based MW measurements characterise only the local scale LWP distributions in the close vicinity of the observational point, and this is their disadvantage if compared to satellite measurements. However they can provide the important information on the diurnal cycle of LWP over land and water surface with high temporal resolution, and also they can be used for validation of satellite data on LWP obtained for the coastline area near the ground-based validation point. The RPG-HATPRO microwave radiometer, which is functioning at the observational site of the Faculty of Physics, St.Petersburg State University (Russia), perfectly suits the requirements to the experiment aimed at the LWP gradient detection. It is located at a distance of 2.5 km from the coastline of the Gulf of Finland and performs angular scanning towards the Gulf of Finland every 20 minutes while doing routine observations.

The idea to use ground-based microwave radiometers in the angular (elevation and azimuth) scanning mode for detecting horizontal gradients and for plotting maps of atmospheric parameters is not new. The 22-channel radiometer MICCY (Microwave Radiometer for Cloud Carthography) with high temporal (1 s) and spatial (antenna beam 1°) resolution and scanning possibilities in horizontal (0-360°) and vertical (0-90°) planes was designed for mapping clouds (Crewell et al., 2001). It should be noted that this radiometer is transportable and can be used for mobile measurements. Another instrument is a 10-channel ASMUWARA, the All-Sky MUlti WAvelength Radiometer. It is a system designed for tropospheric monitoring and it is able to observe the sky in all directions with an angular resolution of 9° (Martin et al., 2006a). Retrieving maps of integrated water vapour and liquid water is one of the purposes of this instrument. The examples of these maps can be found at http://www.iapmw.unibe.ch/research/projects/ASMUWARA/online/, last access: 15 May, 2019. A description of the LWP retrieval algorithm, LWP sky maps and corresponding photographs of the sky are presented in the article by Martin et al. (2006b). A short overview of angular scanning observations of cloud liquid water by ground-based MW radiometers can be found in the article by Westwater et al. (2004). Also, the tomographic approach to the retrieval of LWP should be mentioned which is based on MW observations in angular scanning mode from moving platforms – air-borne and ground-based. This approach was first proposed in the 1980s. Huang et al. (2010) demonstrated the feasibility of tomographically retrieving the spatial structure of cloud liquid water using current microwave radiometric technology and provided several general guidelines to improve future field-based studies of cloud tomography.

It should be mentioned that microwave radiometers are capable to provide the information on the spatial inhomogeneity not only of LWP but also of air humidity. Schween et al. (2011) have shown the potential of a single full-

scanning MW radiometer RPG-HATPRO for detecting horizontal water-vapour variability. They demonstrated that applying a simple linear-gradient model together with an assumed vertical profile derived from the closest radiosonde ascent, the strength and direction of the horizontal-humidity gradient can be determined with a temporal resolution about 15–20 min. Meunier et al. (2015) performed simulated experiments for retrieving two-dimensional water vapor fields using a tomographic approach and multiple ground-based MW radiometers. The goal of the mentioned study was to investigate how the various aspects of the instrument setup (number and spacing of elevation angles and of instruments, number of frequencies, etc.) affected the quality of the retrieved field. Stahli et al. (2011) have proposed an imaging method for both water vapour and liquid clouds which used ground-based observations by the SPIRA ground-based MW radiometer operating at 91 GHz by continuously scanning the sky over a range of elevation angles in a fixed azimuth direction. Marke et al. (2020) studied the influence of a heterogeneous land surface on the spatial distribution of atmospheric water vapor: they used ground-based remote sensing measurements of integrated water vapour (IWV) by a microwave radiometer HATPRO during clear sky conditions at 30° elevation angle (full azimuth scans with 10° step).

While the above mentioned studies considered the general problem of LWP mapping by means of MW observations, the present study deals with the specific task: to assess feasibility of detecting LWP horizontal gradients in the coastline area. We emphasize that the retrieval of LWP over land and water surface in the vicinity of the radiometer and the analysis of an error budget is not the primary goal of our study. In order to get insight into typical qualitative features of the LWP land-sea gradient in the vicinity of the radiometer and to identify the main problems relevant to quantitative analysis of measurements and to the solution of the inverse problem of the LWP retrieval over water area using MW angular scans, we start the investigations by focusing the research on the measurement domain. We examine the results of brightness temperature measurements in several spectral channels of the radiometer and at several elevation angles in order to identify the evidences of the land-sea LWP gradient just in the measured quantity, i.e. MW radiation. The analysis is done for different seasons. To our opinion, such an approach, while being relatively simple, is an efficient way to highlight the main points which require thorough investigation. Nevertheless, we also present some preliminary results of the LWP retrievals over water surface.

To the extent of our knowledge, the studies devoted to the detection of horizontal inhomogeneities of atmospheric parameters from ground-based passive microwave measurements are not numerous and ours is the first attempt to solve the specific problem relevant to the investigation of the LWP gradient in the coastline area. Therefore, we decided that it would be reasonable to present the step-by-step analysis of the problem starting from the consideration of the forward problem and to demonstrate the complexity of the task that faces us. We used the classical approach to the solution of inverse problem of atmospheric optics: analysis of the forward problem on the basis of simulations, analysis of measured quantities for several test cases, tuning the retrieval algorithm, processing the experimental data with the help of this algorithm, and the comparison of the results to the independent data. Although the concept of using angular measurements to characterize water vapor and liquid water path gradients is feasible, its practical applications are very difficult due to the high variability of the

liquid water in the clouds, the inhomogeneity of water vapour, etc.. In addition, we would like to emphasize that the experimental setup of the HATPRO radiometer at our observational site was initially developed for improving temperature retrievals in the lower layers rather than for solving the problem of the LWP gradient detection. However, we managed to apply these measurements to the task under consideration and got promising results.

## 2 Description of the instrument, measurement geometry and data processing algorithm

### 2.1 General formulation of the problem

The 14-channel RPG-HATPRO radiometer (Radiometer Physics GmbH – Humidity And Temperature PROfiler, https://www.radiometer-physics.de/; last access 30 May 2019) is mounted on the top of the metal tower on the roof of the building of the Institute of Physics, St.Petersburg State University, 59.88107°N, 29.82597°E, 56 m a.s.l. The integration time of an instantaneous measurement of atmospheric signal is 1 s. The sampling interval depends on operation mode. In the zenith viewing mode, which is the main observational mode, the sampling interval is about 1-2 s. Every 20 min zenith measurements are interrupted and the angular scanning is done in the North-East direction with the azimuth of 24.7°.

Seven spectral channels located in the 0.5 cm oxygen absorption band (51.26, 52.28, 53.86, 54.94, 56.66, 57.30, 58.00 GHz) provide the information on atmospheric temperature profile, and seven channels located in the centre and the wing of the 1.35 cm water vapour line (22.24, 23.04, 23.84, 25.44, 26.24, 27.84, 31.40 GHz) provide the information on atmospheric humidity profile and cloud liquid water path. Zenith measurements are processed by the multi-parameter retrieval algorithm based on the optimal estimation method (Kostsov, 2015). Previously, the results of LWP retrievals were validated and the error analysis was made (Kostsov et al., 2018a). Zenith and angular measurements in combination are also processed by the built-in quadratic regression retrieval algorithm developed by the instrument manufacturer. Both optimal estimation and regression algorithm independently provide the vertical profiles of temperature, absolute and relative humidity, integrated water vapour, and the cloud liquid water path. It is important to emphasize that the angular scans are used only for temperature retrievals in order to improve the results at the boundary layer altitudes. This is a common procedure for radiometers of this type. The "temperature channels" are optically thick and, as a result, the angular measurements are not affected by horizontal inhomogeneities of atmospheric parameters.

The location of the radiometer with respect to the coastline of the Gulf of Finland (the river Neva bay) is shown in Fig. 1. The distance from the radiometer to the coastline is 2.5 km along the horizontal viewing direction. The horizontal line of sight crosses the opposite coastline of the Gulf of Finland at 18 km distance from the radiometer, and at the 22-26 km distance it passes over the lake Sestroretsky Razliv. The radiometer is located at about 25 km distance from the city centre (St.Petersburg) and at about 50 km distance from the nearest radiosounding station (Voeikovo, WMO ID 26063).

The set of elevation angles of the line of sight of the microwave measurements is the following: 90°, 30°, 19.2°, 14.4°, 11.4°, 8.4°, 6.6°, and 4.8°. The viewing geometry in the vertical plane is shown in Fig. 2. The radiometer is remotely probing the air portions over land at elevation angle 90° and over water areas at 7 elevation angles in the range 4.8°-30°.

Different spectral channels have different response to the spatial distributions of temperature, humidity and cloud liquid water. The channels in the water vapour line and oxygen band (at 22-28 and 51-58 GHz) are mainly influenced by humidity and temperature distributions while the channel in the so-called "transparency window" (31.40 GHz) provides the information on LWP. In order to demonstrate that the "LWP channel" is transparent enough in the entire atmospheric region of interest, we calculated optical depth for this channel along lines of sight corresponding to different elevation angles. The

results are plotted in Fig. 3 as a 2D-map. In order to model maximal absorption, as an input for the calculations we took the profiles of temperature and humidity which are typical for warm and humid days in July in St.Petersburg region. The integrated water vapour was 31 kg m$^{-2}$. The LWP of the modelled cloud was equal to 0.4 kg m$^{-2}$ which is the maximal value for non-rainy clouds. Overcast conditions were modelled; the cloud base and top were selected at 1 km and 2 km correspondingly. One can see that even for this extreme case the optical depth at 31.4 GHz does not exceed 1.8 for the

smallest elevation angle at a horizontal distance of 28 km from the radiometer which is the opposite shore of the Gulf of Finland and about 10 km inland. At the opposite coastline which is 18 km from the radiometer, the optical depth reaches a value of about 1 in its maximum. The obtained results lead to the important conclusion: clouds in the layer 2-4 km over the opposite shore of the Gulf of Finland at about 20 km from the radiometer are detectable at small elevation angles (4.8° – 8.4°). In case such clouds are present, the detection of LWP land-sea gradient for clouds in the lower layers will become

rather complicated task.

The measured atmospheric microwave radiation is registered as a set of brightness temperature values $T_b$ corresponding to observations at spectral channels with central frequencies $\nu$ and elevation angles $\alpha$ and will be designated as $T_{bm}$. Brightness temperature values which are calculated for any given set of atmospheric parameters will be designated below as $T_{bc}$. Data processing was done according to the algorithm which is shown in Fig. 4. The set of $T_{bm}$ is the basic input

to the processing and analysis but zenith and angular observations are treated separately. Zenith observations at all 14 spectral channels are processed by the multi-parameter retrieval algorithm based on the optimal estimation approach. The obtained profiles of atmospheric parameters are then used for calculation of brightness temperature values corresponding to elevation angles of angular scans under the assumption of horizontal homogeneity of the atmosphere. At the next step these calculated values are compared to corresponding measured values. The difference between measured and calculated

brightness temperatures is taken as a main quantity for analysis:

$$D_{TB}(\nu,\alpha) = T_{bc}(\nu,\alpha) - T_{bm}(\nu,\alpha) \tag{1}$$

This quantity can be considered as a sum of several terms:

$$D_{TB}(\nu,\alpha) = D_{grad}(\nu,\alpha) + D_{Tq}(\nu,\alpha) + D_{err}(\nu,\alpha), \tag{2}$$

where $D_{grad}$ is the brightness temperature difference which is directly caused by the difference between LWP of a cloud above the radiometer and LWP of a cloud observed at the elevation angle $\alpha$. For simplicity, this term will be referred below as the LWP gradient signal. $D_{Tq}$ is the brightness temperature difference caused by the horizontal inhomogeneity of temperature and humidity. The term $D_{err}$ is the interfering signal stipulated by errors and uncertainties of different kind. First, we point at the errors in retrieved profiles of atmospheric parameters which are used for calculation of $T_{bc}$ under the assumption of horizontal homogeneity. The contribution of these errors to $D_{err}$ needs more detailed explanation. In order to make this explanation more evident, let us consider the example case with a humidity profile error. Let us imagine the situation when the error (the difference between the true and the retrieved humidity profile) is positive in the lower layers of the troposphere and we know the true profile. If we calculate $T_{bc}$ for zenith direction using the true and the retrieved profile the difference between the obtained $T_{bc}$ values will be small and comparable to the random error of microwave measurements and the $T_{bc}$ value will be very close to $T_{bm}$ value. However, if we calculate $T_{bc}$ for small elevation angles using the "erroneous" profile and compare it to the corresponding $T_{bm}$ value, this difference can be noticeably higher due to the considerable increase of optical path through the layers where the retrieved profile has errors. In our example case, the result would be the overestimation of $T_{bm}$ by $T_{bc}$. Here, one important note should be made: the retrieval errors for profiles have random and systematic components (the latter is caused mainly by a priori information used for retrievals). As a result, the term $D_{err}$ might consist of both components also. The pointing error (elevation angle error) can be another source of $D_{err}$, which is important for small elevation angles. Also, for small elevation angles, the surface emission interference can take place through side lobes of the antenna pattern. When considering small elevation angles, one should keep in mind the uncertainty of refraction calculations stipulated by the uncertainty in the vertical and horizontal distribution of atmospheric humidity.

In order to give an impression of the origin of the LWP gradient signal, in Fig. 5a we present a simplified schematic picture of the MW radiation transfer from the atmosphere to an instrument which makes an observation at some elevation angle. We consider two cases: a cloudy atmosphere and a cloud-free atmosphere (temperature and humidity are assumed to be the same). In the cloudy case, the radiation from cold upper atmospheric layers is considerably absorbed by a cloud, at the same time a cloud itself is a strong emitter of a radiation. As a result, an instrument registers the radiation which is formed mainly in warm atmospheric layers within and below a cloud. In the clear sky case an instrument can "see" upper tropospheric layers which are cold and less dense than the lower layers. Hence in a clear sky case the measured brightness temperature is lower than it is in a cloudy case. This reasoning is valid also in case when clouds over a radiometer and over a water body have different LWP: the lower LWP is, the weaker the emission by cloud and absorption of downwelling radiation are. So the measured brightness temperature for clouds with low LWP will be smaller than for clouds with high LWP.

For characterisation of a magnitude of the LWP gradient signal $D_{grad}$ we present Fig. 5b where we modelled the atmospheric situation with the LWP land-sea difference. According to LWP measurements by the SEVIRI instrument in

2013-2014 in the vicinity of St.Petersburg, the mean LWP over the HATPRO radiometer site was 0.080 kg m$^{-2}$, and the mean LWP over the river Neva bay was 0.040 kg m$^{-2}$ (Kostsov et al., 2018b). We modelled 2D radiative transfer for ground-based measurements using these values and disposing clouds within 1-2 km and 3-4 km altitude layers. The artificial cloud with LWP=0.080 kg m$^{-2}$ was placed over the radiometer location and the artificial cloud with LWP=0.040 kg m$^{-2}$ was placed over the entire water area and over the opposite shore of the Gulf of Finland. Annual mean profiles of pressure, temperature and humidity for St.Petersburg region were taken as a necessary input for calculations and and the assumption of horizontal homogeneity of these parameters was used. Fig. 5b shows that, as expected, the 31.4 GHz channel has the largest LWP gradient signal which reaches 14-16 K for the smallest elevation angle. The signals in the 22.24 GHz and 51.26 GHz channels, which are shown for comparison, do not exceed 6 K. The signal at 51.26 GHz is nearly zero for smallest elevation angle because of its high opacity if compared to other considered channels. For 31.4 GHz and 22.24 GHz channels, the signal is higher when the cloud is located within 3-4 km layer than in case of lower cloud, but this difference is not large (about 2 K).

## 2.2 Modelling of measurements in the atmosphere with scattered clouds

Fig. 5b refers to an overcast atmospheric situation which is the simplest but idealised case for estimation of the magnitude of the LWP gradient effect in the measurement domain. In order to be closer to reality, we simulated the scattered clouds over land and sea in the vicinity of the radiometer using a Monte Carlo method. The observational plane (see Fig. 2) was extended and divided into cells (two rows, each row contained 4 cells of the 12x3.25 km size) located over the Gulf of Finland and two opposite shores. In each cell, the random number generator produced the values of the following cloud parameters: the vertical extent (0.3-2 km, uniform distribution); horizontal size (0.5-5 km, uniform distribution); the cloud placement within a cell (uniform distribution); LWP (lognormal distribution). It should be emphasized that the average horizontal size of generated clouds was much smaller than the size of the water body under investigation. While modelling the LWP values, we considered two situations: one with the existing LWP land-sea gradient and another without such a gradient. The mean LWP values for the first situation were the same as taken previously for overcast conditions: (0.08 and 0.04 kg m$^{-2}$ for land and sea correspondingly). For the second situation, the mean LWP value was taken as 0.08 kg m$^{-2}$ everywhere. The number of generated cases was about 165000. Every instantaneous cloud spatial distribution was combined with one set of the meteoparameter profiles (temperature, pressure, and humidity). For these meteoparameters, the assumption of horizontal homogeneity was used. The sets of profiles were obtained in the course of 2 years of observations by the HATPRO radiometer (2013-2014) with the sampling interval of 2 min. As a result, we obtained a statistical ensemble which characterised all seasons.

The important issue which should be discussed with special attention is the influence of the instrument field-of-view (FOV) on the interpretation of the off-zenith measurements. The 22 and 31 GHz channels are optically transparent even

for small elevation angles. If the vertical distributions of atmospheric parameters within FOV at a certain distance from the radiometer can be approximated by linear functions, the effect of FOV will be negligible. The situation can change crucially in case of scattered clouds, especially small size clouds and small elevation angles. With a 3-degree FOV, the HATPRO radiometer will be sampling an air portion of about 1 km vertical size at 20 km distance from the radiometer. Possible configurations of the observational geometry in case of scattered clouds are illustrated in Fig. A. One can see that small clouds may appear entirely within FOV of the radiometer (as shown in Fig. A for the cloud over the opposite shore). Some clouds may be missed by observations due to their location in between the lines-of-sight (LOS) corresponding to different elevation angles. Two or more scattered clouds may fall into FOV. Moreover, one cloud may be detected both in zenith and off-zenith observations.

Fig. A demonstrates the large variety of atmospheric situations. Obviously, for scattered clouds it makes no sense to compare single zenith and off-zenith observations since the LWP gradient signal is a random value under such conditions. It is evident that taking into account not only the spatial variability of clouds but also their temporal variability, we can speak about the LWP gradient component in measurements only in terms of mean values obtained by averaging over large amount of data. Fig. B presents the statistical distributions of simulated brightness temperatures at 31.4 GHz for four elevation angles. For each angle two situations are considered: one with existing LWP land-sea gradient and another without such gradient. The input data for radiative transfer calculations were the Monte Carlo simulations of scattered clouds described above. One can see from Fig. B that for all angles the distribution "with gradient" is shifted towards smaller brightness temperature values if compared to the distribution "without gradient"; however this effect is less pronounced for the elevation angle 11.4° due to the influence of the clouds over the opposite shore of the water body.

In order to estimate the component in measured quantity, which is related to the LWP land-sea gradient effect, we analyse the difference between the mean values of $T_b$ datasets which were calculated for situations without and with the gradient. This difference is equivalent to the $D_{grad}$ values shown in Fig. 5b and presents a measure of the "useful signal" relevant to the LWP gradient contribution. Therefore, we use the same designation of this difference and show it in Fig. C as a function of the elevation angle. One can see the dramatic contrast to the overcast case (see Fig. 5b). For scattered clouds, there is no increase of the useful signal for smaller elevation angles. Contrariwise, the $D_{grad}$ values for elevation angles 11.4° and 14.4° are lower than for the angles 19.5° and 30°. The sharp decrease of $D_{grad}$ at 11.4° is explained by the influence of high LWP of the clouds over the opposite shore of the water body.

In order to assess if the instrument FOV affects the magnitude of the useful signal, we present in Fig. C the $D_{grad}$ values which were calculated for infinitely narrow beam width, i.e. neglecting FOV. The results show that there are no considerable differences between the cases "accounting for FOV" and "neglecting FOV". One should keep in mind that we compare the results which were obtained by averaging of a very large number of individual measurements.

However the effect of FOV exists and it is illustrated by Fig. D which shows the statistical distribution of the difference between the brightness temperature obtained neglecting FOV and the brightness temperature obtained accounting for FOV. We suggest that this difference is a measure which characterises in the best way the FOV influence on the results of the interpretation of the off-zenith measurements. The effect of FOV exhibits itself in the form of additional measurements noise which has a systematic and a random component. The absolute value of the systematic component (characterised by the mean value of the distribution) is less than 0.5 K for all four considered elevation angles and this value can be considered as negligible. No specific dependence of the systematic component on the elevation angle can be seen. In contrast, the random component, which is characterised by the standard deviation, increases for smaller elevation angles. The obtained values of the random component can be used for the estimation of a minimal number of individual measurements which should be sampled in order to suppress considerably the influence of FOV. For example, for a set consisting of about 600 individual measurements, the random component of the error due to neglecting FOV at the elevation angle 11.4° will be reduced to the value about 0.1 K. It means that for the current experimental setup averaging over the 10 day time period is enough for suppressing the random error due to FOV.

So, the described Monte Carlo simulations of clouds and the brightness temperature calculations lead to several important conclusions. First, we reiterate that for scattered clouds it makes no sense to compare single zenith and off-zenith observations since the LWP gradient signal is a random value under such conditions. Second, for averaged quantities, the magnitude of the component of measured signal determined by the LWP land-sea gradient (useful signal) in case of scattered clouds is rather small and therefore one can expect difficulties in detecting it, especially taking into account the presence of a large number of interfering factors. Third, the instrument FOV affects the results of the off-zenith measurements in case of scattered clouds by introducing additional noise. Its systematic component is small and averaging over several hundred cases can minimise its random component. So the assumption of infinitely small beam width can be used for processing measurements if the analysis is done for averaged quantities.

There is still an emerging question: to what extent the signal relevant to horizontal inhomogeneity of LWP $D_{grad}$ interferes with signals $D_{Tq}$ and $D_{err}$. In order to obtain the most realistic assessment of the magnitude of the latter signals we decided to analyse the results of angular scans which have been made during several cloud-free days, instead of compiling computer models of inhomogeneous temperature and humidity fields suitable for the considered experiment. The obtained estimates are presented in the next section.

**3 Case study**

Forward calculations and their comparisons with measurements are the preliminary and essential steps before solving inverse problems in many studies. Analysis in the measurement domain can be especially useful when considering the multi-parameter inverse problems which physically are ill-posed. The solution of such problems implies the application of a priori

information which can affect the result to a great extent. Besides, in case multiple parameters are retrieved simultaneously, their retrieval errors are coupled in a complex way. These two factors can make the analysis in the domain of sought parameters difficult and ambiguous. Therefore we start with the analysis in the measurement domain for better understanding of the useful and interfering signals. Since clouds are atmospheric objects which are characterised by extremely large spatial and temporal variability and since the experimental setup and geometry were not optimised for considered task, the model simulations should be verified by comparison with experimental data. In addition, the theoretical prediction of the value of useful signal should be compared to the experimental data.

We analysed measurements which were made during different atmospheric situations. These situations were selected on the basis of space-borne measurements of LWP in the vicinity of St.Petersburg by the SEVIRI instrument which had been analysed earlier in the article by Kostsov et al. (2018b). In order to study the parallax effect of the space-borne measurements, Kostsov et al. (2018b) compared the results of LWP measurements made by SEVIRI for two ground pixels: the one which is the nearest to the position of the HATPRO radiometer and the other which is the neighbouring pixel but located over the Gulf of Finland just to the North of the radiometer. Measurements during four days were analysed (6 May 2013, 6 June 2013, 5 October 2014 and 11 October 2014) when large differences between LWP over land and sea were detected. In the present study, the consideration of only two mentioned pixels is not sufficient. When the atmosphere is observed by the radiometer at small elevation angles, the air portions over the opposite shore of the Gulf of Finland will make a contribution to measured radiance. Therefore, the distributions of clouds in pixels 241 and 219 (as shown in Fig. 1) should be taken into account as well as in pixels 243 (the radiometer location) and 242 (the Gulf of Finland). Analysing the SEVIRI LWP data in four pixels, we tried to find the following long lasting atmospheric situations:

A) LWP is equal to zero in all four pixels; a cloud-free atmosphere is everywhere. This situation is best for assessing the $D_{Tq}$ and $D_{err}$ terms in the expression (2).

B) A cloud-free atmosphere is in all pixels except one at the radiometer location. This situation is best for assessing the $D_{grad}$ term in the expression (2) during the most favourable observational conditions (without background signal formed by the clouds over the opposite shore of the Gulf of Finland).

C) A cloud-free atmosphere over water area and clouds over both shores of the Gulf of Finland. This is the worst case for detection of the land-sea LWP gradient since the effect can be masked by the background emission from clouds over the opposite shore.

Prior to analysing the cases, we would like to make a note concerning the accuracy of calculations of the brightness temperature difference. These calculations use the temperature, humidity and cloud liquid water profiles retrieved from zenith observations as an input. It is well known that the ground-based microwave method has rather poor spatial resolution which yields smoothed profiles and the very large uncertainty of the vertical placement of a cloud. This fact is known and it was quantified in a number of studies with the help of DOFS calculation (Degrees Of Freedom for Signal which show the number of independent pieces of information that can be extracted from observations). This essential feature of the transfer

of the downwelling microwave radiation in the considered spectral region exhibits itself both in the forward and inverse problems. The brightness temperature calculations for the zenith and off-zenith geometry are equally insensitive to small scale variations of the parameter distributions along the line of sight. Therefore this smoothing feature does not affect our calculations and relevant conclusions. The current version of the retrieval setup assumes the placement of a cloud inside the 0.5-5.5 km altitude range (low and medium clouds). Outside this range, the cloud liquid water profile is constrained to zero values. The workability of this retrieval setup has been confirmed in the study devoted to cross-validation of different methods of the LWP retrieval (Kostsov et al., 2018a). For liquid water profile, DOFS is less than 2 that means the small influence of the liquid water distribution on the results of the brightness temperature calculations. This fact indicates implicitly that the placement of the cloud does not play a crucial role in forward calculations and in the solution of the inverse problem. Also, a kind of proof for that is a wide use of regression algorithms for joint IWV (integrated water vapour) and LWP retrieval from 2-channel observations under the conditions of large uncertainty of the temperature profile and without any information on the cloud vertical location. Based on the above mentioned reasons, we consider the applied radiative transfer model accurate enough for making comparisons between measured and calculated brightness temperature values. Also, it is important to note that most of the cases which were selected for analysis are characterized by clear sky conditions over the water area, therefore the cloud placement error is absent for the off-zenith calculations.

In Fig. 6 the LWP values detected by SEVIRI in four measurement pixels are displayed as a function of time for the date 25 August 2013 (warm and humid season). Accordingly, the values of brightness temperature difference $D_{TB}$ for the set of elevation angles are plotted in the form of 2D time charts for two spectral channels. The colour scale contains 3 parts. The pure yellow part corresponds to the brightness temperature difference in the interval [-1 K; 1 K]. An appearance of yellow colour in a 2D plot means that the difference between measurement and model calculation is negligibly small for corresponding combination time/elevation angle. The red hue describes positive values of $D_{TB}$, the blue hue describes negative values. Fig. 6 refers to a cloud-free atmospheric situation as detected by SEVIRI instrument: the LWP values are all equal to zero except for pixel 219 after 267.7 fractional day, however those values are less than 0.008 kg m$^{-2}$ and can be considered as negligibly small. Here and below we use the UTC for time scales and fractional days. The day count starts on 1 December 2012 – the first day of selected datasets. Local noon is at 0.416 day fraction (11:00 UTC). One can see that for the 31 GHz channel $D_{TB}$ values are close to zero for the elevation angle 30°. For smaller elevation angles, $D_{TB}$ becomes negative and its absolute value increases. The map has only one specific signature: at about 267.2 fractional day the absolute value of negative $D_{TB}$ is the largest reaching 14 K and 26 K for 31 GHZ and 22 GHz channels correspondingly. In general, the brightness temperature difference for the 22 GHz channel is noticeably larger than for the 31 GHz channel. The reason for that is the larger optical thickness of the 22 GHz channel and higher sensitivity of this channel to water vapour variations.

Fig. 7 is similar to Fig. 6 and also refers to cloud-free conditions but during cold and dry season (2 March 2013). In contrast to 25 August 2013, the results for 31 GHz channel demonstrate negligibly small difference between measured and calculated brightness temperature within the whole range of elevation angles. Some negative values appear occasionally at

elevation angles below 10°. For the 22 GHz channel, the difference between measured and calculated brightness temperature is negligibly small within the range of elevation angles 10°-30°. For lower angles, $D_{TB}$ becomes negative, but its absolute values are not large. This case is an example of a very small influence of the humidity variations on $D_{Tq}$ and $D_{err}$ in the 31 GHz channel in a dry atmosphere.

The next plot (Fig. 8) corresponds to the case 1 May 2013 which is the combination of the mentioned above atmospheric situations A and B. It is very important to note that it would be wrong to directly compare the signatures in the LWP plot (a) and in the 2D time charts for $D_{TB}$ (b) and (c). The LWP of the SEVIRI retrieval is the result of averaging over the area of about 7x7 km while measurements by the HATPRO radiometer are very local. In contrast to the study (Kostsov et al., 2018b) in which only zenith observations with frequent data sampling were used, we can not perform averaging of

HATPRO measurements because sampling interval for angular scans (20 min) was quite large for that. This fact should be taken into account when comparisons of (a) panel with (b) and (c) panels are made. One should not expect the precise agreement of signatures on a time scale. Fig. 8 demonstrates large number of positive values of $D_{TB}$ for the 31 GHz channel. The largest of them reach 4 K and correspond to the period of time when SEVIRI detected clouds over the ground-based radiometer (about 151.3-151.4 fractional day). These positive values observed for all elevation angles are the LWP land-sea

gradient signal which is perfectly seen in the considered case despite the fact that it is not large and does not exceed 4.5 K. For the cloud-free part of the day (starting approximately from 151.45 fractional day) we see the appearance of negative $D_{TB}$ values with the largest absolute brightness temperature difference at small elevation angles. For the 22 GHz channel, negative $D_{TB}$ were detected at small elevation angles all day long.

Let us consider the most interesting case which is described by Fig. 9. This is the case with heavy cloudiness (LWP is

reaching 0.3 kg m$^{-2}$) over both shores of the Gulf of Finland and clear conditions over water area (25 July 2013). We stress, that we have the information on the spatial distribution of clouds only from the SEVIRI observations. Unfortunately, the ground-based measurements for 25 July 2013 are available starting only from 236.34 fractional day, nevertheless the observational period is long enough for analysis. First, we point at the large amplitude of the brightness temperature difference: from -18 K to 24 K. The reason for that is the presence of clouds with high LWP. Second, we point at the mixture

of positive and negative $D_{TB}$ values for 31 GHz channel within the time period 236.34-236.6 fractional day. As it was already noted, the ground-based measurements are very local, instantaneous and not averaged. Therefore, if the cloud distribution is fragmented, the disposition of separate clouds over the radiometer, over water area and over the opposite shore of the Gulf of Finland may be considered to a certain extent random. This fact manifests itself as a mixture of positive and negative $D_{TB}$. As a result, the LWP land-sea gradient, which obviously existed during the considered day according to

SEVIRI observations, is completely masked due to presence of cloudiness over the opposite shore of the Gulf of Finland. Starting from 236.6 fractional day, clouds disappeared everywhere and for this period the $D_{TB}$ 2D map is more homogeneous. Similar to cloud-free situations during warm and humid season described by Figs. 6 and 8, the $D_{TB}$ values are

predominantly negative for this period and the absolute difference of brightness temperatures is larger for small elevation angles.

Fig. 10 illustrates atmospheric conditions similar to Fig. 8 but the LWP values of the clouds over the radiometer are much larger (up to 0.25 kg m$^{-2}$). At the same time there are some clouds with much smaller LWP over the opposite shore of the Gulf of Finland. We see that for the 31 GHz channel positive $D_{TB}$ prevail showing the evidence of considerable LWP land-sea gradient even for small elevation angles. For the 22 GHz channel, in contrast to Fig. 8, the $D_{TB}$ are also predominantly positive even for small elevation angles. The reason for that is high signal originating from the clouds with

large LWP. The "separation of variables" in channels 22 GHz and 31 GHz is obviously not perfect, that is why the 22 GHz channel is also sensitive to cloud liquid water, as 31 GHz channel is sensitive to humidity distribution. As a result, in the considered case the positive signal of the LWP land-sea gradient $D_{grad}$ dominates in the 22 GHz channel over the negative values of the sum of the terms $D_{Tq}$ and $D_{err}$ (especially for small elevation angles).

        Concluding this section, we can formulate the following statements:

1)  As predicted, the LWP land-sea gradient (higher LWP over land, lower LWP over water) is detectable and shows up as positive values of the difference between modelled and measured brightness temperatures of the MW radiation. These positive values can be seen in the whole considered range of elevation angles (4.8°-30°). The experiment revealed that the magnitude of the useful signal ($D_{grad}$) can vary from 2 K to 24 K depending on elevation angle and LWP land-sea difference (as it is provided by the SEVIRI satellite instrument). Obviously, thorough quantitative analysis is

problematic due to the fact that the true state of the atmosphere over the water body (the Gulf of Finland) was unknown: the SEVIRI instrument provided averaged data on LWP, and there was no information on corresponding pressure, temperature, humidity profiles and type of cloudiness.

        2)  The effect of LWP land-sea gradient can be masked by the signal from clouds over the opposite shore of the Gulf of Finland.

3)  There is a systematic negative component of the brightness temperature difference $D_{TB}$ which is clearly revealed under cloud-free conditions and can reach in the warm and humid season 20K by its absolute value at small elevation angles. So far, we do not have enough information for accurate identification of the origin of this negative component. Pointing error (elevation angle systematic error) should have produced signal which is constant in time, so it is not the case. The uncertainty of accounting for refraction is smaller by more than the order of magnitude. The interfering signal coming

from the surface through side lobes of the antenna pattern is very unlikely to be the reason since the effect depends on air humidity. So, the only two explanations remain: the humidity horizontal gradient or the amplification of the systematic error of humidity retrieval when brightness temperatures are calculated for elevation angles other than 90°. The presence of the negative component of $D_{TB}$ can make it difficult to detect LWP land-sea gradients if these gradients are not very pronounced.

## 4 Statistical characteristics: seasonal features

The main idea of this statistical analysis is to compare the monthly mean values of two quantities: $D_{LWP}$ and $D_{TB}$. Here, $D_{LWP}$ is the difference between LWP obtained by SEVIRI in pixels 243 (land, radiometer location) and 242 (sea, Gulf of Finland) and this quantity in our study is the reference measure of the LWP land-sea gradient. $D_{TB}$ is the brightness temperature difference in the 31.4 GHz channel which has been defined in section 2 and contains the component reflecting the LWP land-sea gradient.

In order to minimise the influence of the interfering systematic negative component of $D_{TB}$ attributed to the humidity horizontal gradient, in statistical analysis we consider only the elevation angles larger than 10°. The other advantage of this limitation is the missing of most clouds over the opposite shore of the Gulf of Finland, over second small water area (Sestroretsky Razliv) and the land at about 28 km distance, because the atmospheric layers below approximately 4 km are not scanned. For the sake of correct comparison of the ground-based and space-borne measurements we omitted all HATPRO and SEVIRI measurements made for solar zenith angle (SZA) larger than 72° since the retrieval errors of the LWP measurements by SEVIRI strongly increase for the large SZA. The SEVIRI and HATPRO data sets used for calculations of monthly mean values contained all available high quality measurements. The elements of these data sets were not synchronised, which means for example that when HATPRO did not produce the data because of rain or snow, the SEVIRI data set might have had no gaps.

The monthly mean values of $D_{LWP}$ and $D_{TB}$ are plotted in Fig. 11 separately for 2013 and 2014 for warm and cold seasons. Prior to discussion of Fig. 11 two important notes should be made. First, due to presence of the systematic component (interfering signal) originating, as suggested, from the horizontal inhomogeneity of water vapour, the attention should be paid to the qualitative temporal behaviour of $D_{TB}$ rather than to the specific values of this quantity. And second, one should account for possible influence of seasonal variation of the interfering systematic component on the temporal dependence of $D_{TB}$.

As one can see from Fig. 11a, the LWP gradient detected by the SEVIRI instrument during the WH season has two maxima (in May-June and in October) and one minimum in August-September. Comparing $D_{LWP}$ and $D_{TB}$ for the WH season we note similar temporal behaviour of these quantities within the time interval May – August. The best agreement is observed for 2014. For 2013 the agreement is not as good as in 2014 since the ground-based measurements demonstrate profound minimum in June which is not present in the satellite measurements. For the CD season, there is a good agreement of temporal behaviour of $D_{LWP}$ and $D_{TB}$ in 2013: maxima in February and April and minimum in March. There is no agreement for the CD season of 2014: the satellite data show slight decrease of the LWP gradient within time interval February-April while the ground-based data show its increase. There is one interesting feature that should be also noted: the monthly mean values of $D_{TB}$ for different elevation angles are very close to each other for all seasons. However, the variability of $D_{TB}$ in 2013 at small elevation angles (11° and 14°) is higher than for large elevation angles (19.2° and 30°).

It should be reiterated that both water vapour and cloud liquid water affect the brightness temperature values which are registered in the so-called "humidity channels" (22 – 31 GHz, K-band). When we analyse Fig. 11, we keep in mind the interfering influence of atmospheric humidity on the values of $D_{TB}$. In order to perform a separation of variables in our problem, we need to abandon the analysis of the quantities in the measurement domain (brightness temperatures) and to start the analysis in the domain of sought parameters which in our case are LWP and IWV (integrated water vapour). The simplest and commonly used method to solve the inverse problem of the LWP and IWV retrieval from microwave observations in the K-band of microwave spectra is the application of regression algorithms – linear or quadratic. Both algorithms have advantages and disadvantages; therefore we decided to apply both of them and to compare the results. The regression formulae for the LWP value are as follows:

$$LWP_n = \sum_{k=1}^{L} a_{kn} T_{kn} + a_{(L+1)n} \qquad (3)$$

$$LWP_n = \sum_{k=1}^{L} b_{kn} T_{kn} + \sum_{k=1}^{L} b_{(L+k)n} T_{kn}^2 + b_{(2L+1)n} \qquad (4)$$

where Eq. (3) refers to linear regression, Eq. (4) refers to quadratic regression; $n$ identifies the elevation angle of observations, in our case $n=0,…,7$ (zero refers to zenith viewing); $a$ and $b$ are the regression coefficients, index $k$ identifies the spectral channel, $L$ is the total number of spectral channels which are considered in the regression scheme; $T$ is the brightness temperature. In the present study, we used for retrievals only two of seven spectral channels in the K-band: 22.24 GHz and 31.40 GHz, so $L=2$ in Eqs. (3) and (4).

In the course of developing the retrieval algorithm, we used two variants of training data sets. At first, we trained the algorithm separately for each of the seasons and years and considered only the overcast case with limited range of variations of the cloud base and the cloud vertical extension. This approach appeared to be ineffectual and did not produce robust results. It was found that extensive forward modelling of scattered clouds with highly variable parameters was necessary. Therefore, finally, training of the regression algorithms was performed on the basis of the Monte Carlo modelling of the atmosphere with scattered clouds described in subsection 2.2. The complete training dataset included the values of LWP calculated along the line-of-sight and converted to the LWP in the vertical column. In case of crossing several clouds by the line-of-sight the LWPs from all these clouds were taken into account. The brightness temperatures at 22.24 GHz and 31.40 GHz were calculated accounting for the instrument FOV. This training dataset was used to derive the regression coefficients. As a result, for each of the regression algorithms (linear or quadratic) of the LWP retrieval we had at our disposal 8 sets of regression coefficients corresponding to 8 elevation angles. Testing of the regression algorithms in the numerical experiments conducted for simulated overcast conditions and scattered clouds has shown that the algorithms overestimate the true LWP for off-zenith observations with the bias in the range 0.003-0.006 kg m$^{-2}$ (for elevation angle 60°). The bias slightly increases for smaller elevation angles. For zenith observations, the bias is negligibly small. So, we can make the conclusion that the algorithms can not overestimate the LWP gradient, if it is detected while processing field measurements.

After applying the regression algorithms to the brightness temperature values measured at different elevation angles we could estimate the land-sea LWP difference as obtained from ground-based MW observations using the formula:

$$D_{Hn} = LWP_0 - LWP_n \tag{5}$$

where $n$ stands for elevation angle and zero refers to zenith viewing as it was in Eqs. 3 and 4, the index H indicates that the data refer to HATPRO. The results of estimation of the land-sea LWP difference both by space-borne and ground-based observations are presented in Fig. 12. This plot is organised similar to Fig. 11, but contains only one vertical axis ($D_{LWP}$). The results obtained by linear and quadratic algorithms appeared to be very similar, so we present the results of the linear algorithm only.

Prior to analysis of Fig. 12, several preliminary remarks should be made. First, in order to exclude possible rainy conditions from the satellite data we removed all LWP greater than 0.4 kg m$^{-2}$ from the SEVIRI dataset before plotting Fig. 12. The second remark concerns possible influence of the clouds over the opposite shore of the Gulf of Finland on the results of the estimation of the land-sea LWP difference from ground-based observations. In order to make a proper comparison of ground-based and satellite data for such a situation, we have calculated the land-sea LWP difference from the SEVIRI data using three different formulae:

$$D_{S1} = LWP_{243} - LWP_{242} \tag{6}$$

$$D_{S2} = LWP_{243} - \left(LWP_{242} + LWP_{241}\right)/2. \tag{7}$$

$$D_{S2} = LWP_{243} - \left(LWP_{242} + LWP_{241} + LWP_{219}\right)/3. \tag{8}$$

Eq. 6 corresponds to pure land-sea LWP gradient which is estimated as the difference between LWP for the land and sea pixels. Eq. 7 models the situation when the HATPRO instrument is probing air portions over sea and over the opposite coastline of the Gulf of Finland for medium elevation angles. The "sea value" of LWP in this case is combined from the equal contributions by pixels 242 (sea) and 241 (opposite coastline are). And Eq. 8 is intended for modelling the HATPRO observations at small elevation angles. In this case there can be an additional contribution from clouds inland relatively far from the opposite coastline, i.e. over pixel 219. Again, as for the previous case, the contributions of pixels to the "sea value" of LWP are equal.

We would like to emphasize that the extensive and thorough comparison of the HATPRO and SEVIRI data on LWP for pixel 243 has already been made and the results have been published (Kostsov et al., 2018b, 2019). Good agreement for daily mean LWP of the ground-based and satellite data has been revealed. Moreover, the cross-comparison of the HATPRO LWP data with the data from two space-borne instruments SEVIRI and AVHRR confirmed the agreement not only for averaged values, but also for single measurements (Kostsov et al., 2019). To date, there were no attempts to compare the satellite and ground-based data on LWP over water surfaces. However, the validity of the satellite data over large water

bodies was confirmed implicitly by the comparison of the SEVIRI and AVHRR results over the Gulf of Finland and the Lake Ladoga (Kostsov et al., 2019).

Taking into account the remarks made above, we can analyse Fig. 12. First of all, we pay attention to the fact that after removing the LWP values greater than $0.4 \, \text{kg m}^{-2}$ from the SEVIRI datasets the $D_{LWP}$ derived from satellite observations became much smaller than shown in Fig. 11 for the complete datasets. However the temporal behaviour remains the same as in Fig. 11 for all seasons if we look at $D_{S1}$. If we look at $D_{S2}$ and $D_{S3}$ we can notice the increase of values from February to March 2013 instead of decrease as shown in Fig. 11. The most important result shown in Fig. 12 is

that the ground-based microwave measurements definitely detect the LWP land-sea gradient during all seasons and this gradient is positive as in case of the satellite measurements (larger LWP values over land and smaller over sea). The gradient is negative only for March 2013 but its corresponding absolute value is small. Comparing the gradients obtained by the ground-based measurements during warm and cold seasons we may conclude that in general the gradients during cold season are smaller than during warm season and not as variable as during warm season. For warm season, the gradient derived from

microwave measurements at the 60° elevation angle is smaller than the gradients obtained from measurements at other elevation angles. It is interesting to note that there are no noticeable differences between the values corresponding to elevation angles 11.4°, 14.4° and 19.2° during warm season and between the values corresponding to all considered angles during cold season. This fact leads to the conclusion that the clouds over the opposite shore do not produce a noticeable influence on the results. Therefore hereafter when comparing the SEVIRI and HATPRO data we shall consider only the $D_{S1}$

values.

        For the warm seasons of 2013 and 2014, temporal behaviour of the LWP gradient revealed by the satellite measurements completely differs from that obtained by the ground-based measurements. The satellite measurements show two local maxima in June-July and in October while the ground-based measurements demonstrate maxima in May and August-September. The maximal values of the gradient derived from satellite observations are much larger than the maximal

values of the gradient derived from ground-based observations. In contrast to the warm season, during the cold season the temporal behaviour of the gradient is the same for the SEVIRI and the HATPRO results. In order to find any explanations for the agreement of the results in terms of temporal behaviour during cold season and the disagreement during warm season, additional investigations are necessary involving thorough assessment of the error budget of the results – not only ground-based but also derived from satellite observations. It should be noticed that the analysis of the quantities in the measurements

domain demonstrated several similar patterns in temporal behaviour of $D_{TB}$ and $D_{LWP}$ during warm season of 2014 and cold season of 2013.

        It is interesting to compare the obtained values of the LWP land-sea gradient with the data which are provided by reanalysis, namely ERA-Interim from ECMWF (Dee et al., 2011). The main shortcoming of such comparison is the coarse spatial resolution of the reanalysis data. The internal resolution of the ECMWF data is 0.75 deg, i.e. about 80 km which is

too poor to describe the scene of our experiment. For higher resolutions of the reanalysis data, the interpolation procedure is applied, but the highest recommended resolution is 0.25 deg (28 km). So we have chosen the 28 km resolution but even in this case we could not apply the reanalysis data to the scene of our experiment. Therefore we selected two areas 0.25×0.25 deg which are the nearest to the HATPRO radiometer and which represent the land surface and the water body. The location of these areas on a map is shown in Fig. E. The ECMWF data for land surface refers to the territory located

about 30 km to the south from the HATPRO radiometer. The ECMWF data for the water surface refers to the territory located about 120 km to the west and 30 km to the north from the measurement site. The ECMWF data on LWP for 6 and 12 UTC were collected and averaged over a period of one month.

The comparison of the LWP gradient from SEVIRI, HATPRO and the ECMWF reanalysis is presented in Fig. F. Due to large displacement of the reanalysis data we can not expect the agreement in temporal behaviour but we can compare the

average magnitude of the LWP gradient. For a warm season, one can see a very good coincidence of the magnitude of the LWP gradient derived from the ground-based observations and provided by reanalysis. The best agreement can be seen for the period May-July/August. The discrepancies increase during the period August-October 2014. For the cold season in contrast to SEVIRI and HATPRO, the reanalysis provides negative LWP land-sea gradients. However, the absolute values of these gradients are not large. The HATPRO results display positive gradients and the temporal patterns are similar to the

patterns shown by the SEVIRI data. In general, we can make three main conclusions from this comparison. First, the SEVIRI and the HATPRO instruments detect positive LWP land-sea gradients during all seasons but the magnitude of the gradient detected by the ground-based instrument is considerably smaller than detected by the satellite instrument. Second, the LWP gradients provided by HATPRO and reanalysis during the warm season are in a very good agreement. Third, the reanalysis data demonstrate negative LWP gradient during cold season in contrast to the SEVIRI and the HATPRO data. The

mean values of the LWP land-sea gradient for all considered time periods are given in Table T1. One can see that there are no noticeable seasonal differences in the SEVIRI data while the HATPRO results demonstrate lower values during cold season. The analysis of physical reasons for the seasonal differences in the LWP land-sea gradient is beyond the scope of the present study. To our opinion, such analysis requires much more data including the satellite data sampled over various water bodies.

Also, Fig. F demonstrates how some factors affect the obtained results. We present $D_{LWP}$ obtained by the HATPRO instrument at the elevation angle 14.4° for three scenarios of training the regression algorithm. The main scenario describes scattered clouds, existing LWP land-sea gradient, and the microwave measurements with the account for FOV. The second scenario neglects FOV and the third one describes the conditions without LWP land-sea gradient. One can see both factors produce negligibly small effect on the obtained results. The conclusion was expected since neglecting FOV is equivalent to

the presence of additional random noise which is suppressed by averaging. Also, it is important to mention that the presence of the LWP land-sea gradient in the training data set does not automatically provide its detection when processing the field

campaign data. The training was performed with respect to LWP values rather than the gradient values. Besides, the training was performed for each elevation angle separately.

## 5 Discussion and identification of problems

### 5.1 Data sampling

Data sampling issue seems to be of primary importance for the solution of the problem of detecting the land-sea LWP gradient. In our case, the angular scan is performed every 20 min. This time interval is very large for cloud studies. Rose et al. (2005) has noted that the integration time (or sampling interval) should not be greater than 20 s in order to register the short-period variations of tropospheric humidity and cloud liquid water. Kostsov et al. (2016) have estimated the optimal value of sampling interval of ground-based microwave observations by HATPRO using the information approach: the values of the information volume calculated for measurement sequences with different sampling intervals have been compared. The integration time was always the same and equal to 1 s, the lower sampling rates were obtained by sparsely sampling the data. The sampling interval that corresponded to the maximum of the information volume was considered as optimal. Kostsov et al. (2016) have made the conclusion that even for stable atmospheric situation the sampling interval should not be greater than 100–200 s. In this case maximum information could be extracted form MW measurements.

For detection of land surface induced atmospheric water vapour patterns, Marke et al. (2020) used passive MW measurements by the HATPRO radiometer in zenith direction and in azimuth scanning mode at the elevation angle of 30°. The interval between scans varied from 10 to 30 min. This interval is similar to the interval in our study. However, it should be specially noted that Marke et al. (2020) investigated only clear sky cases without any considerable advection.

Taking into account the above mentioned findings relevant to the sampling interval studies we can conclude that the shortest possible sampling interval would be the best solution. The clouds are a highly variable atmospheric object. The problem of detection of the LWP gradient can be considered as an estimation of a small difference of two large quantities. These quantities are the LWP values over land surface and water body. The solution of the problem requires simultaneous and frequent measurements of these quantities which are variable in space and time. Obviously, the problem can not be solved without averaging of measurements over specific time periods. The long averaging periods and the short sampling intervals are preferable for obtaining accurate estimates of the LWP gradient. The value of 10 s for sampling interval seems to be the optimal trade-off: the short-period variations can be registered keeping the amount of data not very large. However, the angular scanning procedure itself consumes some time: for HATPRO, one angular scan takes 4.5 min. Thus, several practical suggestions can be made, for example:

- to implement scan-by-scan observational mode with small number of elevation angles in order to increase the sampling rate, in this case the sampling interval could be shortened to 1-2 min;

- to alternate 20 min period of zenith observations with 20 min period of observations at one selected elevation angle and to use the sampling rate of 10 s within these periods.

These suggestions could be helpful also with respect to the problem of comparison of the ground-based and satellite data. Such a comparison requires time averaging of the ground-based data. Different studies recommend different time periods and weighting functions for averaging. Our experience (Kostsov et al., 2018b, 2019) has shown that the period of 20 min is a good choice for comparisons with the data delivered by satellite instruments SEVIRI and AVHRR.

## 5.2 Orientation of the instrument

It has been shown by the case study (see section 3) that clouds over the opposite shore of the Gulf of Finland can play an interfering role and mask the effect of the LWP land-sea gradient in angular observations. Fig. 2 demonstrates geometrically that clouds located over the opposite shore in the altitude layer 2-4 km can be detected by observations at three smallest elevation angles. The lake Sestroretsky Razliv located not far from the opposite coastline is a small water body (see Fig. 1). Therefore one can not expect strong influence of this water body on cloud properties, and the entire area within 18-28 km distance along horizontal projection of the line of sight can be assumed as "land".

If we look at both Figures 1 and 2 we can come to the conclusion that the optimal orientation of the horizontal projection of the line of sight could be strictly to the North. In this case the line of site would pass the long distance (up to about 30 km) over the Gulf of Finland which is the main water body in our research. The interfering influence of clouds over the opposite shore of the Gulf of Finland would be minimized. At the same time the line of sight would not pass over the island Kotlin which can be a source of heat as a land surface and as an urban area (the city of Kronstadt occupies part of the island territory). However it should be noted that the HATPRO instrument operating at St.Petersburg University is firmly attached to the metal tower and has no appliance for rotation azimuthally, so changing its orientation requires special actions.

## 5.3 Data processing algorithm

In the present study we considered only one algorithm of the derivation of LWP from microwave observations which was based on regression relationships linking measured brightness temperature values and LWP. The regression algorithm (linear or quadratic) is widely used for processing the microwave observation data. Simplicity and computational efficiency are its main advantages. The other algorithm is called "physical" or "physical-iterative" and it is based on the inversion of the radiative transfer equation, usually by optimal estimation method (Rodgers, 2000). The detailed analysis of the applicability of both algorithms and of their combination to the problem of derivation of LWP and integrated water vapour (IWV) from two-channel microwave observations was done by Turner et al. (2007). In general, the superiority of the physical algorithm over regression algorithm originates from the fact that this method accounts for the spatial distribution of all parameters which influence the radiative transfer in the considered spectral channels. In particular, the information about temperature in

cloud layers helps to reduce the LWP retrieval errors. The applicability of the physical method to the problem of the LWP and IWV retrieval by two-channel radiometers implies that the a priori profiles of pressure, temperature and humidity are available from external data sources and the cloud liquid water profile is assigned in a model form. In the process of solving the inverse problem by the physical method, cloud liquid water and humidity profiles are modified in one way or another to deliver minimum to the residual between measured and simulated brightness temperatures. For multi-channel radiometers, all mentioned profiles, including temperature and pressure ones can be derived from microwave observations simultaneously. Also, the microwave measurements can be combined with other measurement data and constraints. Such approach is called IPT (integrated profiling technique) or general approach to solution of multi-parameter inverse problems (Loehnert et al., 2008; Kostsov, 2015ab).

Since the physical approach is more accurate than the regression approach its application to the considered problem of the detection of LWP land-sea gradient seems to be a promising direction of a further research. One should keep in mind that measurements at different elevation angles are treated separately due to horizontal inhomogeneity of atmospheric parameters. Therefore the considered inverse problem in its general formulation through the radiative transfer equation will be the classical strongly underdetermined ill-posed problem which will require a system of constraints.

## 5.4 Systematic component of signal

Last but not least we discuss the systematic component which was detected in measured brightness temperature. First of all, we note that when azimuth scans at different elevation angles are performed the directional dependent interference can be present in measured signal. For example, Marke et al. (2020) registered such interference in the unprotected 26.24 GHz channel at four specific azimuth directions. Corresponding measurements were filtered out and missing values were filled with linear interpolation. In our case, we can not determine whether the systematic component is directionally dependent or not, since there is no possibility to perform azimuthal scanning (the radiometer is firmly attached to the stand and has no appliance for the azimuthal rotation). In Section 3 we have made the statement that so far we do not have enough information for accurate identification of the origin of the negative component of brightness temperature in the water vapour channel and the LWP channel of the radiometer. However, there is a high probability that this component reflects the horizontal gradient of the air absolute humidity. If this hypothesis is accepted, then we have to explain the origin of high absolute humidity over the Gulf of Finland and/or over the territory located between the radiometer and the Gulf of Finland. High content of water vapour over the water body can be explained either by the evaporation or by the advection of humid air. Considering the problem of the quantification of evaporation from lakes Finch and Calver (2008) note, in particular, that:

− There are a number of factors that can affect the evaporation rates; first of all, one can mention the climate and physiography of the water body and its surroundings. Also the stored heat can be transported within the water body itself and into and out of it.

– Seasonal variations in the evaporation rate depend on the heat storage capacity of the water body which is greatly determined by its depth.

– Seasonal variations of the evaporation rate are not necessarily synchronised with seasonal variations of the net solar radiation; as the water depth increases, the maximum evaporation can be observed within the period from one to four months after the summer solstice.

– The significant factor influencing the evaporation rate is the heat which is transferred into a water body by inflows and outflows. The variety of inflows includes seepage from groundwater bodies, changes in bank storage, rivers flowing into

the water body and land surface run off. Enumerating outflows, one can mention rivers, controlled withdrawals (reservoirs) and leakage to groundwater.

The Neva bay, the part of the Gulf of Finland over which the line of sight of the radiometer passes, is very shallow, its depth does not exceed several meters. The Neva bay is separated from the main part of the Gulf of Finland by the dam. Therefore, to a first approximation, the Neva bay may be considered as a big lake with the Neva River as the major inflow. The

exchange of water between the Neva bay and the main part of the Gulf of Finland goes on through several special passages in the dam. Taking into account all factors presented above, one can suggest that investigation of the seasonal behaviour of the systematic component would be reasonable action in order to attribute it to the evaporation from the Neva bay.

The land surface territory between the radiometer and the nearest coastline of the Gulf of Finland can be also a source of evaporation. This territory is covered by the forest (park). In the study by Marke et al. (2020) devoted to land surface

induced atmospheric water vapor patterns, it has been shown that less water vapour seems to be present at elevated deciduous forest. In our case the forest is not elevated, however one can not expect a pattern of extra humidity over the forest.

The systematic component of the brightness temperature can be caused not only by high absolute humidity along the line of sight but also by the larger air temperature than expected under the approximation of the temperature horizontal

homogeneity. The measured signal is affected by air temperature directly through the emission of radiation and indirectly through the temperature dependence of the absorption by water vapour and liquid water. The line of sight at elevation angles other than 90° passes in its horizontal projection about 150 m over the roof of the building of the Institute of Physics which can be a kind of heat source, especially during sunny days when the roof is warmed up. In addition, there should be an air temperature gradient over the coastline itself. These factors can also contribute to systematic component of signal.

**5.5 Measurement geometry and data quality control**

When the HATPRO measurements in the zenith direction are processed routinely, the data quality control procedure includes several steps. The first step is filtering out the data obtained during rain events (as detected by the rain sensor) and during a certain period after a rain event. The duration of this period is taken equal to 4 hours as recommended in the special

study (Kostsov et al., 2018a). At the next step the convergence of the iterative process of the inversion of the radiative transfer equation is analysed. The convergence limit is set to 12 iterations. All data corresponding to unconverged processes are filtered out. It should be noted that normally the number of iterations before successful convergence varies from 5 to 9. The last check refers to the analysis of the residual between measured brightness temperature values and the corresponding values calculated on the basis of the retrieved atmospheric parameters. In case the RMS residual exceeds 1 K, the results are considered erroneous. This 3-step procedure helps to keep only the good quality data.

Measurement geometry which is used and analysed in the present study is based on angular scanning. Such geometry gives the possibility to probe remotely the air portions which are located very far from the radiometer in the horizontal direction. In this case a situation may occur when the line of sight passes through a rain event (a shower) while there is no rain at the radiometer location and the rain sensor detects no rain. When the regression algorithm is used for the LWP retrieval, it is difficult to ensure the sufficient data quality control. However, the application of the physical method (already discussed in section 5.3) would allow implementing the described above second and third steps of quality control procedure similar to the case with zenith observations.

There is another aspect relevant to the measurement geometry which should be mentioned. The solution of the problem of the detection of the LWP land-sea gradient implies the combination of zenith and angular microwave observations. While zenith observations deliver the absolutely local "spot" data over the radiometer (the horizontal dimension is determined by the beam width), the data obtained at angular observations may be considered as averaged over a certain horizontal distance. For small elevation angles this distance can reach dozen of kilometres. If we take into account the high temporal and spatial variability of clouds, the direct comparison of the results of zenith and angular observations made during one scan can be erroneous. Probably, more rigorous way of comparison would require temporal averaging of the results of zenith observations over a certain period of time as it is done, for example, when ground-based measurements of LWP are compared to the satellite data. The satellite data are spatially averaged over a ground pixel area and in order to perform proper comparison the ground-based data are time averaged over a period approximately equal to a time of an air parcel movement at a given wind speed through a ground pixel of a satellite measurement. For the problem which is considered in the present study, one could suggest performing zenith measurements with high sampling rate and the subsequent averaging of them just before making an angular scan.

**6 Summary and conclusions**

Previously, the measurements of the cloud liquid water path (LWP) by the SEVIRI and AVHRR satellite instruments provided the evidences of the systematic differences between LWP values over land and water areas in Northern Europe. In the present study an attempt is made to detect such differences by means of ground-based microwave observations performed near the coastline of the Gulf of Finland in the vicinity of St.Petersburg, Russia. The microwave radiometer

RPG-HATPRO located 2.5 km from the coastline is functioning in the angular scanning mode and is probing the air portions over land (at elevation angle 90°) and over water area (at 7 elevation angles in the range 4.8°-30°). The data obtained within the time period December 2012 – November 2014 were taken for analysis.

In this study we used the classical approach to the solution of inverse problem of atmospheric optics: analysis of the forward problem on the basis of simulations, analysis of measured quantities for several test cases, tuning the retrieval
algorithm, processing the experimental data with the help of this algorithm, and the comparison of the results to the independent data. The decision to make such step-by-step analysis was stipulated by the fact that although the concept of using angular measurements to characterize water vapor and liquid water path gradients is feasible, its practical applications are very difficult due to the high variability of the liquid water in the clouds, the inhomogeneity of water vapor, etc.. The high temporal and spatial variability of cloud parameters (vertical and horizontal placement, horizontal size, LWP, vertical
extension) are the reason for solving the problem of detection of the LWP land-sea gradients only on the basis of averaging of a large number of measurements.

At the first stage on the basis of simulations including the Monte Carlo simulations of the atmosphere with scattered clouds, the assessment was done of the magnitude of the LWP land-sea gradient signal in the brightness temperature measurements. The estimations show that the mean value of this signal at 31.4 GHz can vary in a wide range from 2.5 K for
scattered clouds up to 4-14 K for overcast conditions. The instrument field-of-view (FOV) affects the results of the off-zenith measurements in case of scattered clouds by introducing additional noise. The systematic component of this noise is small and averaging over several hundred cases can minimise its random component. So the assumption of infinitely small beam width can be used for processing measurements if the analysis is done for averaged quantities.

At the second stage of investigations the problem of the LWP gradient detection is examined in the measurement
domain in the special case study. The brightness temperatures of the microwave radiation measured at different elevation angles in the 31.4 GHz and 22.24 GHz spectral channels are analysed and compared with the corresponding values which were calculated under the assumption of horizontal homogeneity of the atmosphere. The difference between measured and calculated brightness temperatures $D_{TB}$ is taken as a main quantity for analysis. Several specific cases, selected on the basis of the satellite observations by the SEVIRI instrument were considered in detail including: clear-sky conditions, the presence
of clouds over the radiometer and at the same time the absence of clouds over the Gulf of Finland, and the overcast conditions over the radiometer and over the opposite shore of the Gulf of Finland. As predicted, the LWP land-sea gradient (higher LWP over land, lower LWP over water) shows up as positive values of the difference between modelled and measured brightness temperatures of the MW radiation. The analysis of the test cases revealed that the magnitude of the LWP gradient signal in brightness temperature measurements can vary from 2 K to 24 K depending on elevation angle and
LWP land-sea difference (as it is provided by the SEVIRI satellite instrument). These positive values can be detected in the whole considered range of elevation angles (4.8°-30°). The effect of LWP land-sea gradient at small elevation angles can be

masked by the signal from clouds over the opposite shore of the Gulf of Finland. Besides, there is a systematic negative component of the brightness temperature difference which is clearly revealed under cloud-free conditions and can reach in the warm and humid season 20K by its absolute value at small elevation angles. So far, we do not have enough information

for accurate identification of the origin of this negative component.

The analysis of monthly mean values of $D_{TB}$ at 31.4 GHz (the LWP gradient signal in the measurement domain) does not lead to unambiguous conclusion about the existence of the LWP land-sea gradient since the sign of these values is alternating. However, several similar patterns were detected in the temporal behaviour of $D_{TB}$ and the LWP gradient derived from the satellite observations by the SEVIRI instrument (in particular for May-August of 2013 and 2014 and for February-

April 2013). The presence of these similar patterns confirmed the conclusion that the systematic component in measurements makes the analysis in the brightness temperature domain (i.e. measurement domain) complicated. The suggestion has been made that this systematic component is caused by water vapour inhomogeneity. In order to perform a separation of variables in our problem, we abandoned the analysis of the quantities in the measurement domain and started the analysis in the domain of sought parameters. Linear and quadratic regressions have been selected as suitable retrieval algorithms for the

LWP retrievals.

Training of the regression algorithms was performed on the basis of the Monte Carlo modelling of the atmosphere with scattered clouds which was used for extensive simulations of the microwave measurements when the forward problem was analysed. In the present study, we used for retrievals only two of seven spectral channels in the K-band: 22.24 GHz and 31.40 GHz. Testing of the regression algorithms in the numerical experiments conducted for simulated overcast conditions

and scattered clouds has shown that the algorithms overestimate the true LWP for off-zenith observations with the bias in the range 0.003-0.006 kg m$^{-2}$ (for elevation angle 60°). The bias slightly increases for smaller elevation angles. For zenith observations, the bias is negligibly small. So, we can make the conclusion that the algorithms can not overestimate the LWP gradient, if it is detected while processing field measurements. The linear and quadratic regression algorithms produced similar results, therefore the results obtained by the linear regression algorithm only are presented in the article.

The most important result is that the LWP retrievals definitely demonstrate the existence of the LWP land-sea gradient during all seasons and this gradient is positive as in case of the satellite measurements (larger LWP values over land and smaller over sea). The gradient is negative only for March 2013 but its corresponding absolute value is small. Comparing the gradients obtained by the ground-based microwave measurements during warm and cold seasons we may conclude that in general the gradients during the cold season are smaller than during the warm season and not as variable as

during the warm season.

The intercomparison of the LWP land-sea gradient data from the HATPRO and SEVIRI measurements and the ECMWF reanalysis has been carried out. The SEVIRI and the HATPRO instruments detect positive LWP land-sea gradients

during all seasons but the magnitude of the gradient detected by the ground-based instrument is considerably smaller than detected by the satellite instrument. For the warm seasons of 2013 and 2014, temporal behaviour of the LWP gradient revealed by the satellite measurements completely differ from that obtained by the ground-based measurements. In contrast to warm season, during cold season the temporal behaviour of the gradient is the same for the SEVIRI and the HATPRO results. The LWP gradients provided by HATPRO and reanalysis during warm season are in a very good agreement. During cold season in contrast to the SEVIRI and the HATPRO data, the reanalysis data demonstrate negative LWP gradient.

The main conclusion of the study is the following: the approach to detection of the land-sea LWP gradient from microwave measurements by the HATPRO radiometer operating at the observational site of St.Petersburg State University has been successfully tested and the results confirmed the presence of the horizontal land-sea LWP gradient in the vicinity of the radiometer. Further research is needed in order to increase the accuracy of the retrieval method and to find the explanations for the revealed differences in the magnitude and temporal behaviour of the LWP gradient obtained from the ground-based, satellite and reanalysis data. The study has identified several problems: sparse data sampling in angular scanning mode, not optimal azimuthal orientation of the instrument, the necessity to improve the data processing algorithm and the need to find the origin of the systematic component in signal measured in angular scanning mode.

**Data availability**

The LWP data derived from the RPG-HATPRO observations at the measurement site of Saint Petersburg State University are available upon request (please write to Vladimir Kostsov at v.kostsov@spbu.ru). The LWP data derived from the SEVIRI observations are available at https://www.cmsaf.eu, last access: 15 May 2019 (EUMETSAT CM SAF, 2019).

**Author contributions**

VSK conceived the study, made the cloud liquid water path retrievals from ground-based microwave observations and prepared the draft of the manuscript. DVI and AK were in charge of the satellite data analysis. VSK, DVI and AK together interpreted the results, reviewed and edited the manuscript.

**Competing interests**

The authors declare that they have no conflict of interest.

**Acknowledgements**

The operation of the RPG-HATPRO instrument was provided by the Research Centre GEOMODEL of St. Petersburg State University (http://geomodel.spbu.ru/). The authors thank E. Yu. Biryukov for the Monte Carlo simulations of the atmosphere

with scattered clouds. The authors are grateful to two anonymous referees for making very insightful remarks and for introducing several useful ideas which helped greatly to improve the manuscript.

**Funding**

This research has been supported by Russian Foundation for Basic Research through the project No. 19-05-00372.

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

Table T1. Mean values of the LWP land-sea gradient (kg m$^{-2}$) for different time periods derived from the SEVIRI and the HATPRO observations and provided by the ECMW reanalysis.

| Season | SEVIRI | HATPRO | ECMWF |
|--------|--------|--------|-------|
| 2013WH | 0.022 | 0.011 | 0.009 |
| 2014WH | 0.025 | 0.013 | 0.006 |
| 2013CD | 0.018 | 0.003 | -0.005 |
| 2014CD | 0.022 | 0.005 | -0.003 |

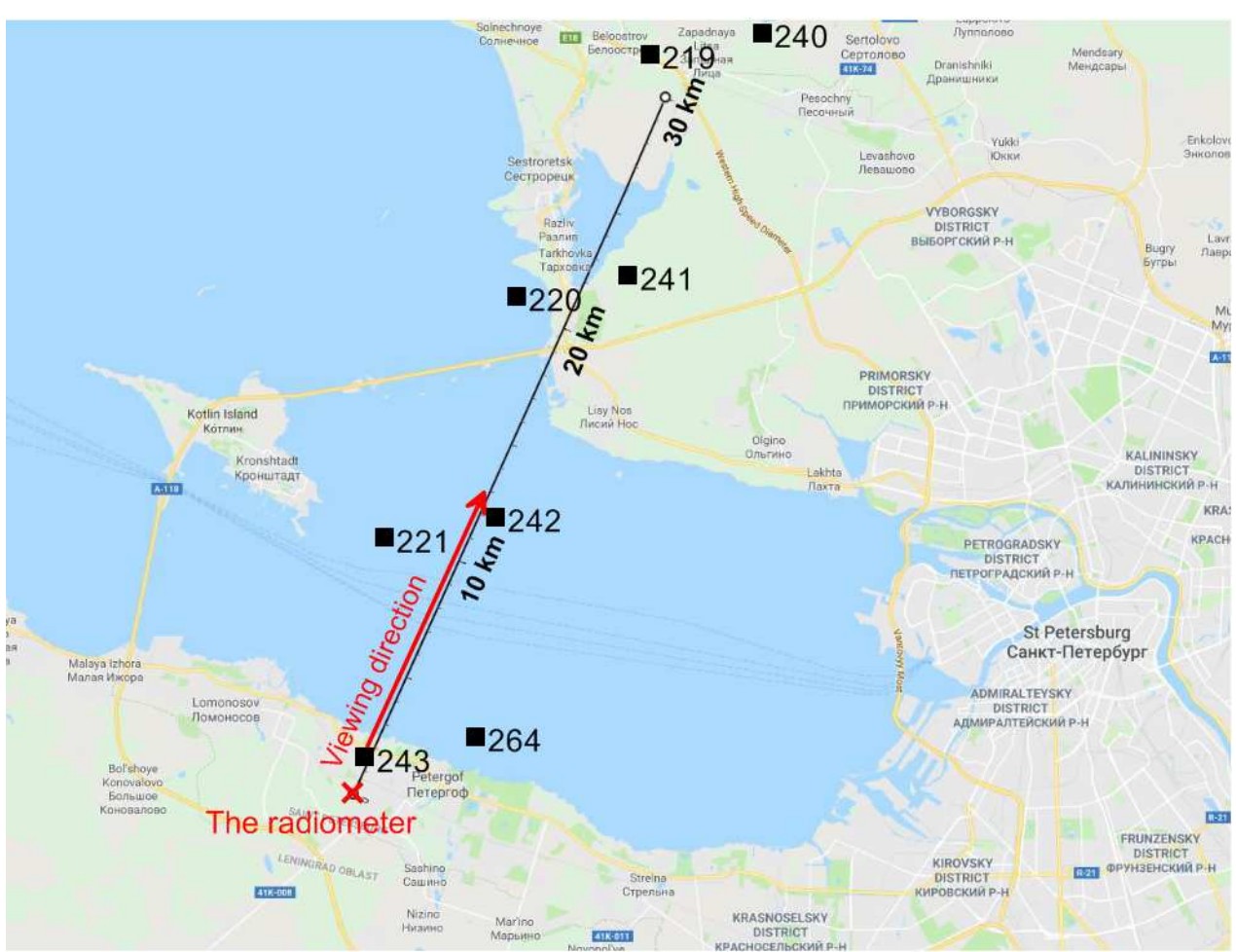

**Figure 1: The location of the RPG-HATPRO radiometer and the viewing direction in the angular scanning mode. The black straight line is the distance scale. Black squares (with numbers) show the position of the centres of the SEVIRI measurement pixels. Map data ©2019 Google.**

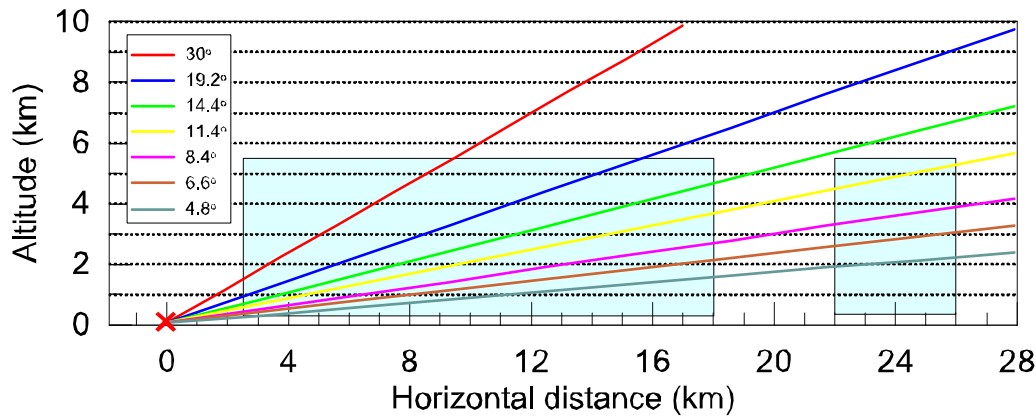

**Figure 2: The viewing geometry in the vertical plane. Position of the radiometer is marked by the red cross. Colour lines represent the lines of sight for different elevation angles (see the legend). Blue boxes designate the atmospheric layer 0.3-5.5 km over water areas (see text).**

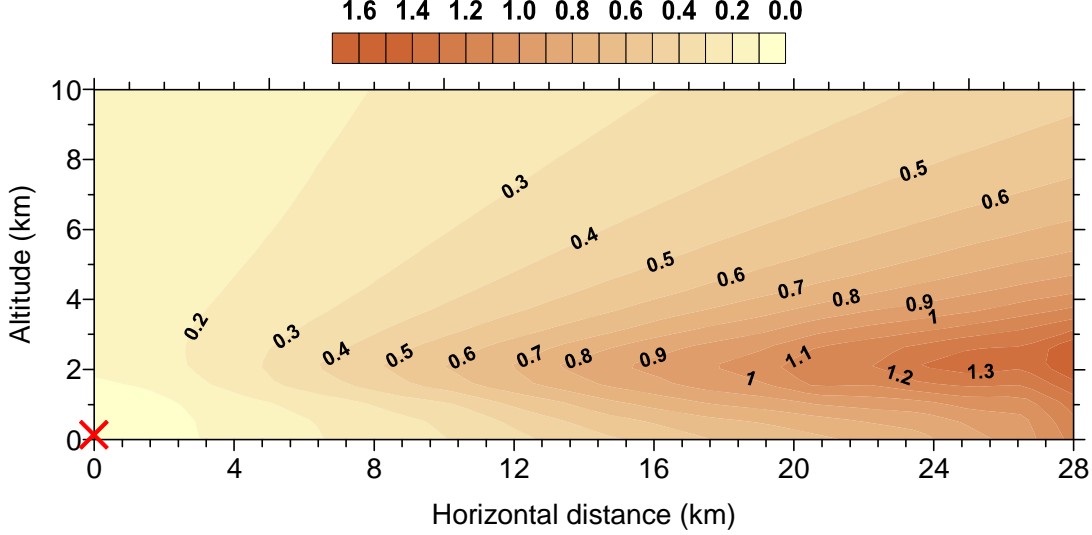

**Figure 3: The 2D distribution of optical depth in the 31.4 GHz channel as calculated from the radiometer location point (marked by the red cross). Overcast conditions, cloud base is 1 km, cloud top is 2 km, LWP=0.4 kg m$^{-2}$.**

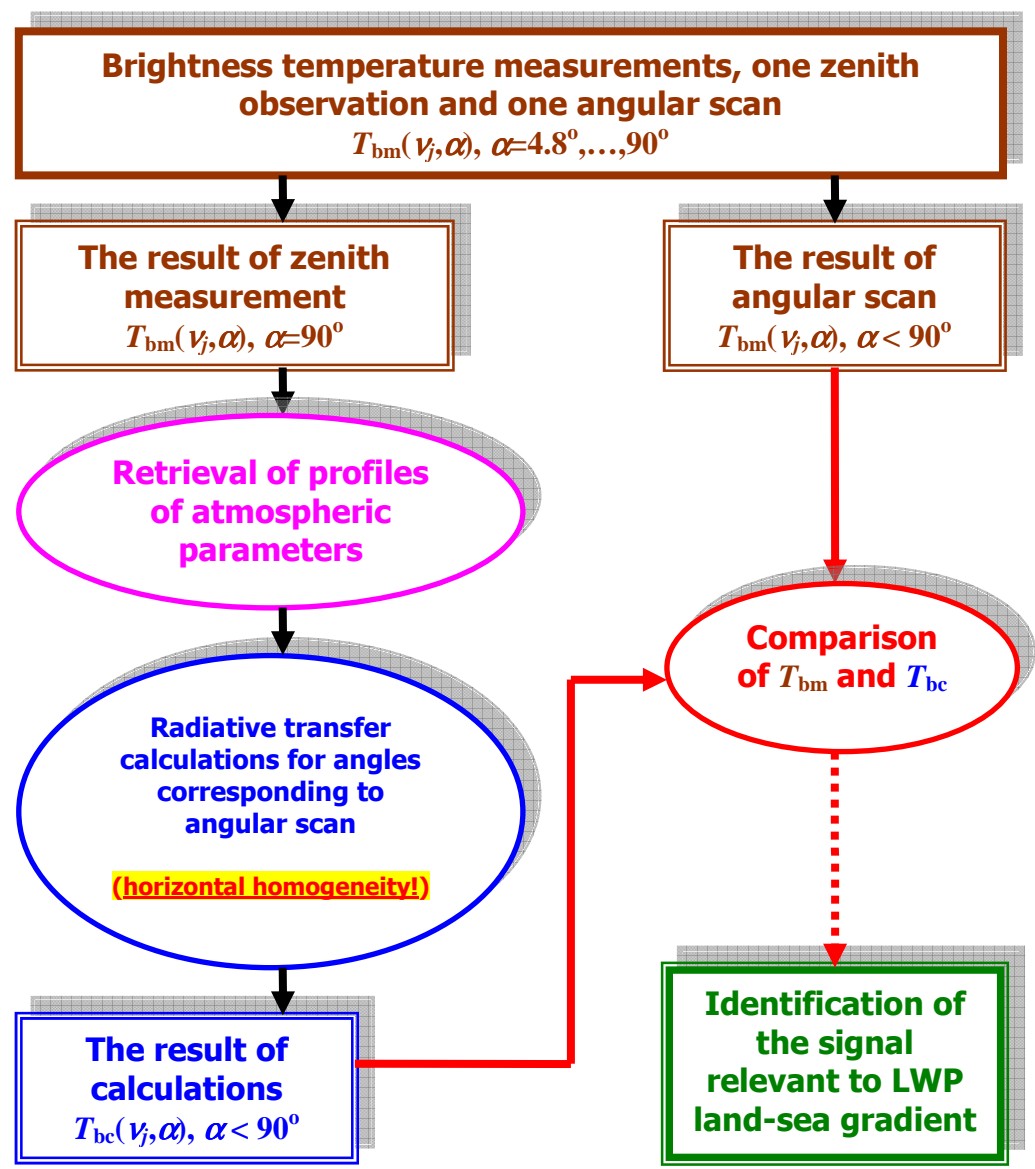

Figure 4: The algorithm for data processing and analysis.

(a)

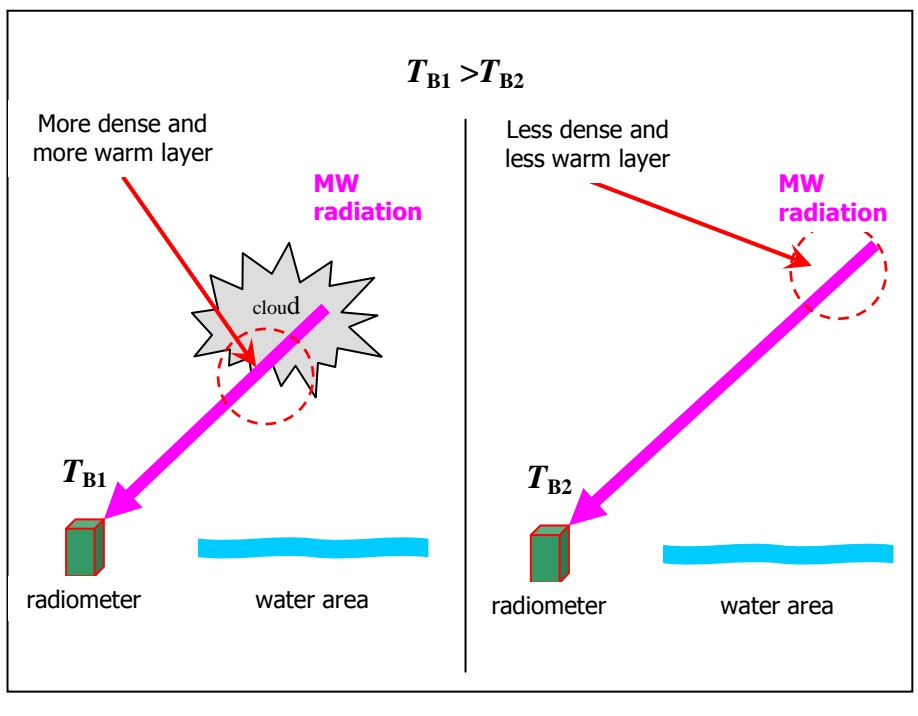

(b)

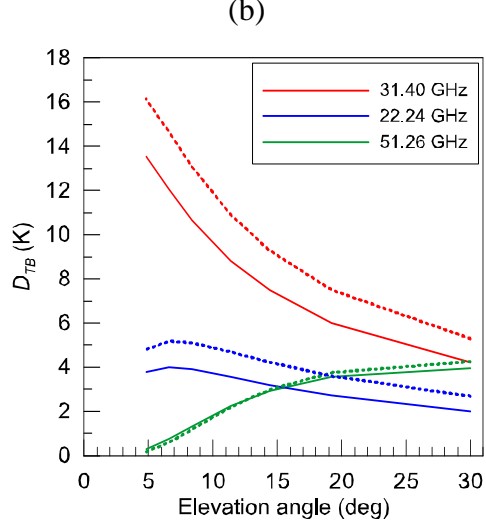

**Figure 5: (a) A simplified scheme of the MW radiation transfer from the atmosphere to an instrument illustrating the origin of the LWP gradient signal. (b) The LWP gradient signal $D_{grad}$ as a function of the elevation angle in three spectral channels. $LWP_{land}=0.080$ kg m$^{-2}$, $LWP_{sea}=0.040$ kg m$^{-2}$. Solid and dashed lines correspond to the cloud located within 1-2 km and 3-4 km layers correspondingly.**

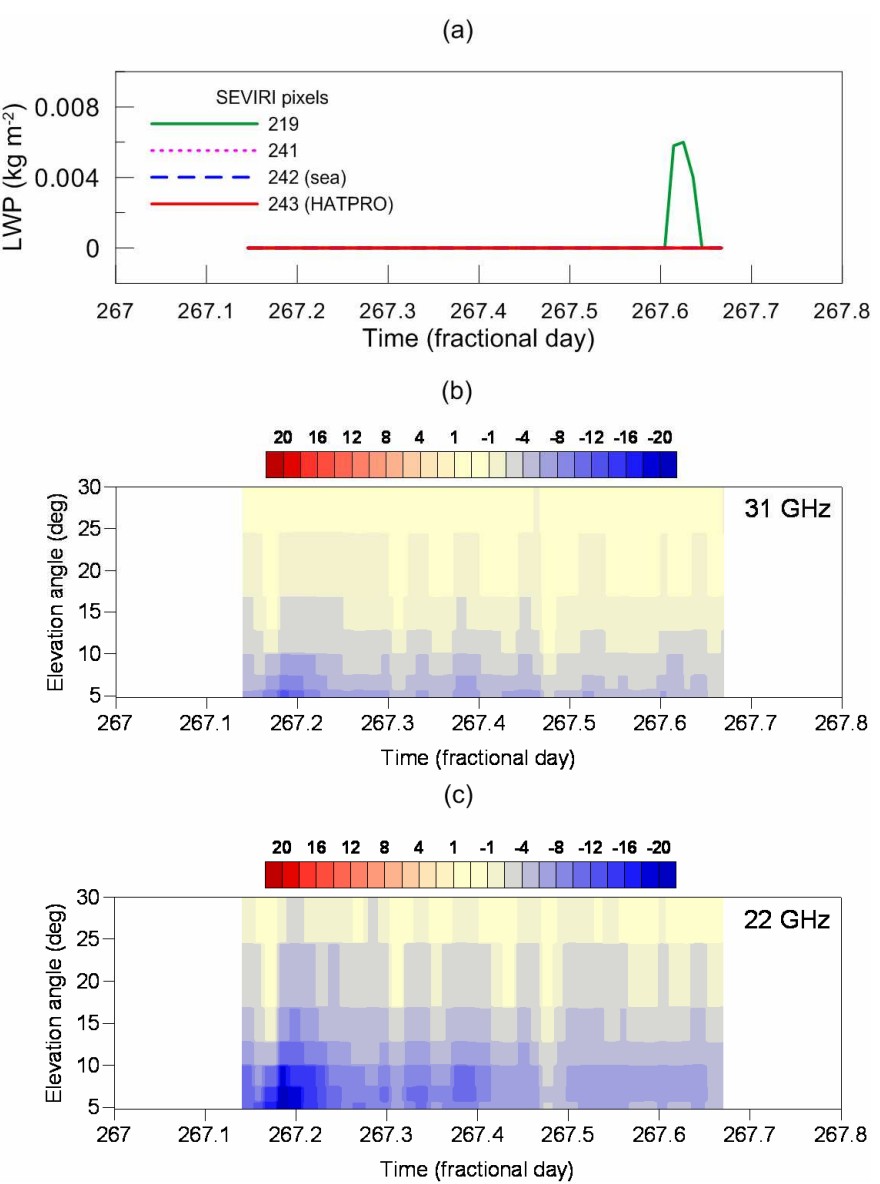

**Figure 6: (a) The LWP in four SEVIRI pixels as a function of time. (b,c) The difference between calculated and measured brightness temperatures DTB (colour scale) as a function of time and elevation angle for two spectral channels (2D plots, the channel frequency is indicated in the plots). 25 August 2013.**

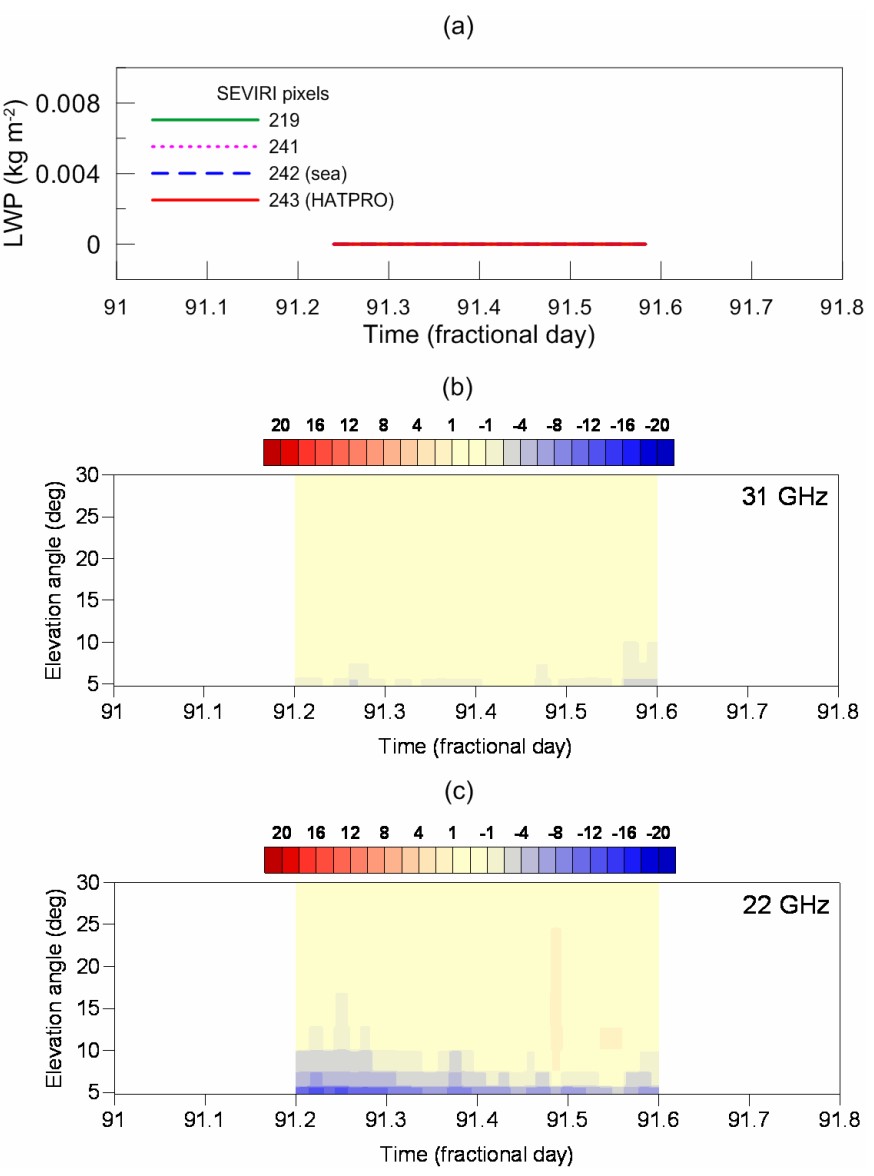

**Figure 7: The same as Fig. 6 but for 2 March 2013.**

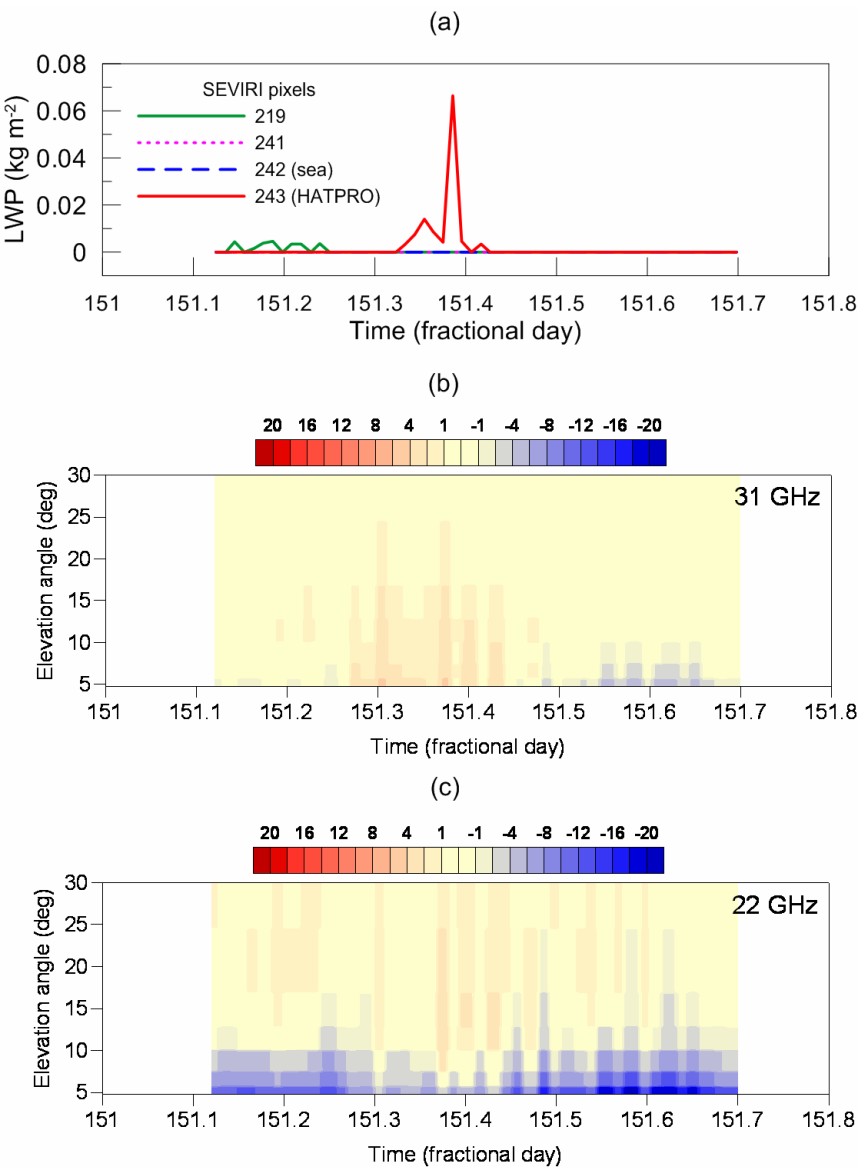

**Figure 8: The same as Fig. 6 but for 1 May 2013.**

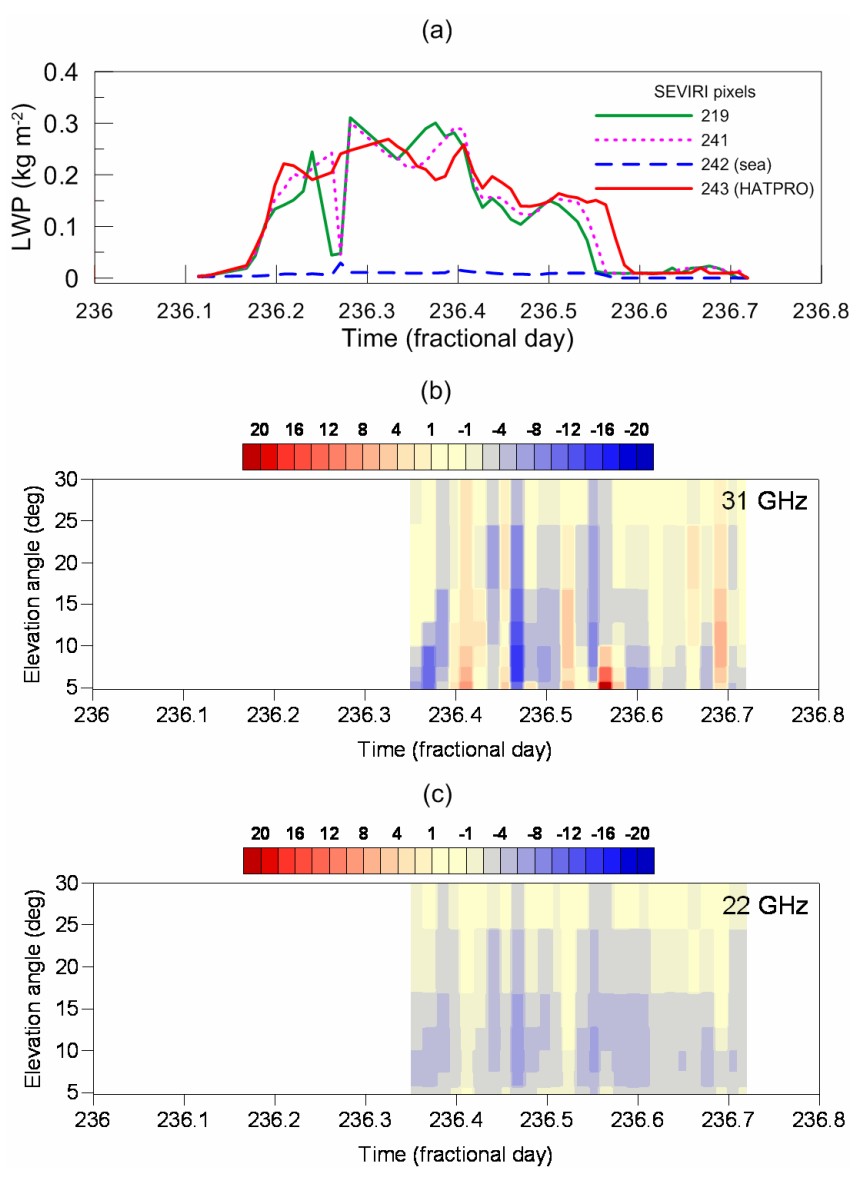

**Figure 9: The same as Fig. 6 but for 25 July 2013.**

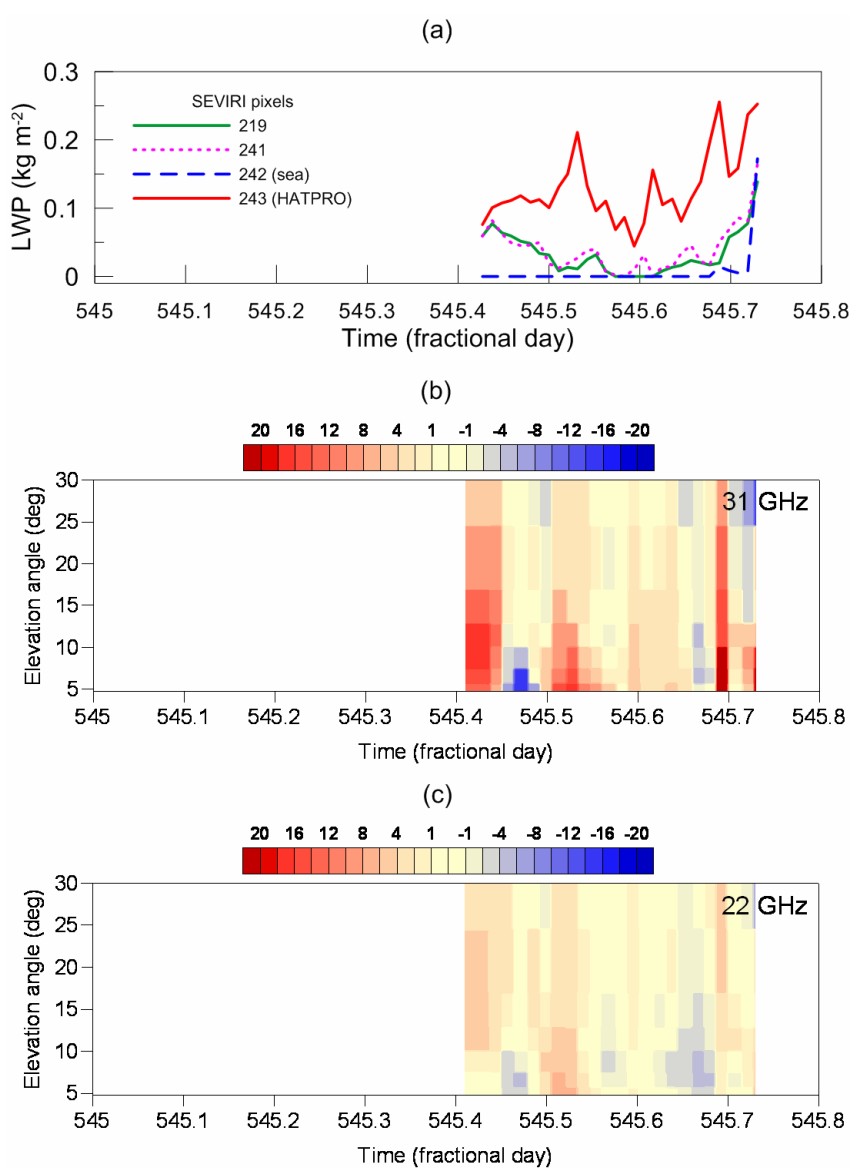

**Figure 10: The same as Fig. 6 but for 30 May 2014.**

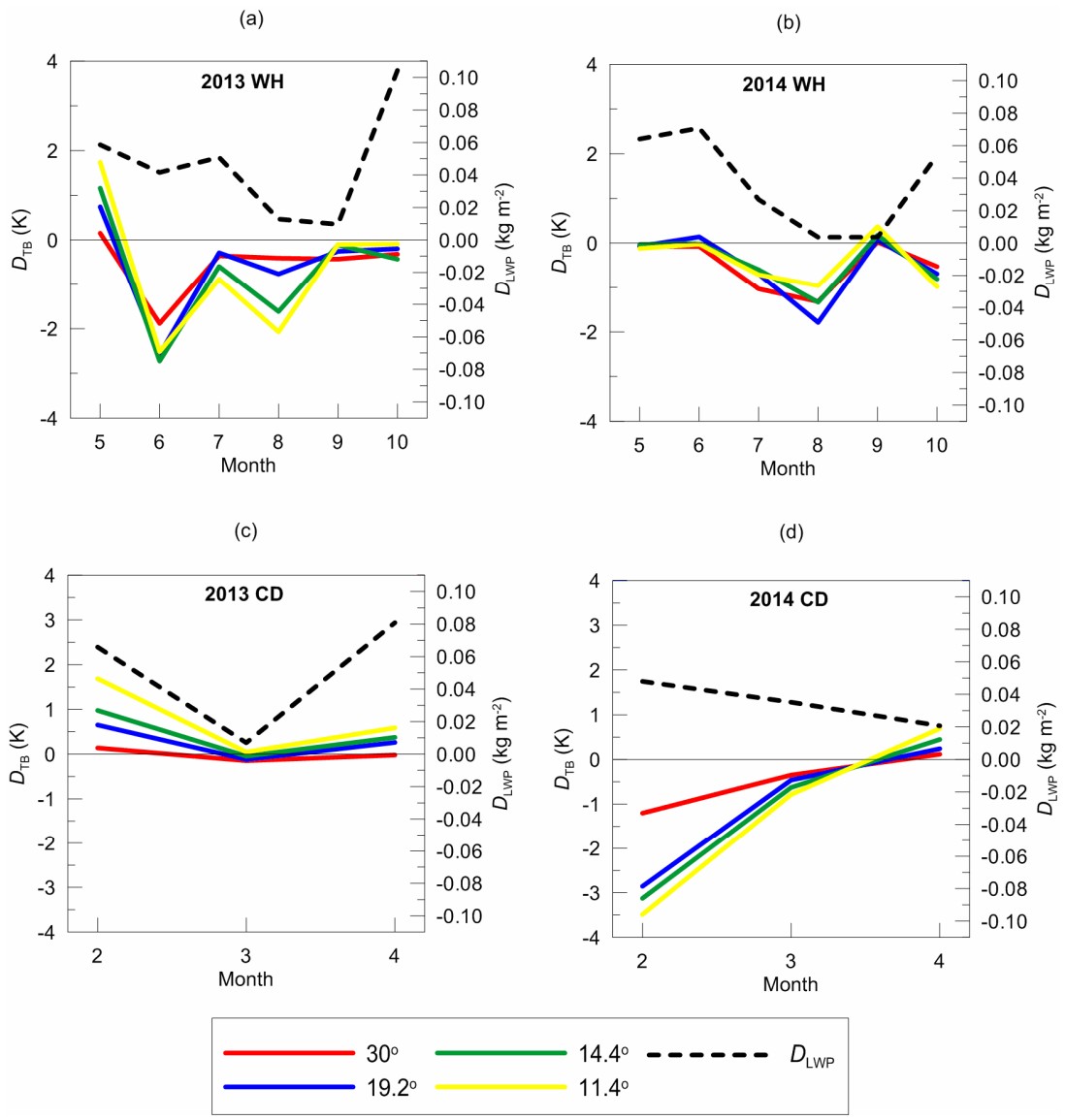

**Figure 11: Monthly mean brightness temperature difference $D_{TB}$ (left y-axis, colour lines correspond to different elevation angles, see the legend) and monthly mean LWP land-sea difference $D_{LWP}$ (right y-axis) as functions of time for warm and humid season of 2013 (a) and 2014 (b) and for cold and dry season of 2013 (c) and 2014 (d). $D_{LWP}$ is defined as LWP(243) minus LWP(242) where the numbers denote SEVIRI ground pixels.**

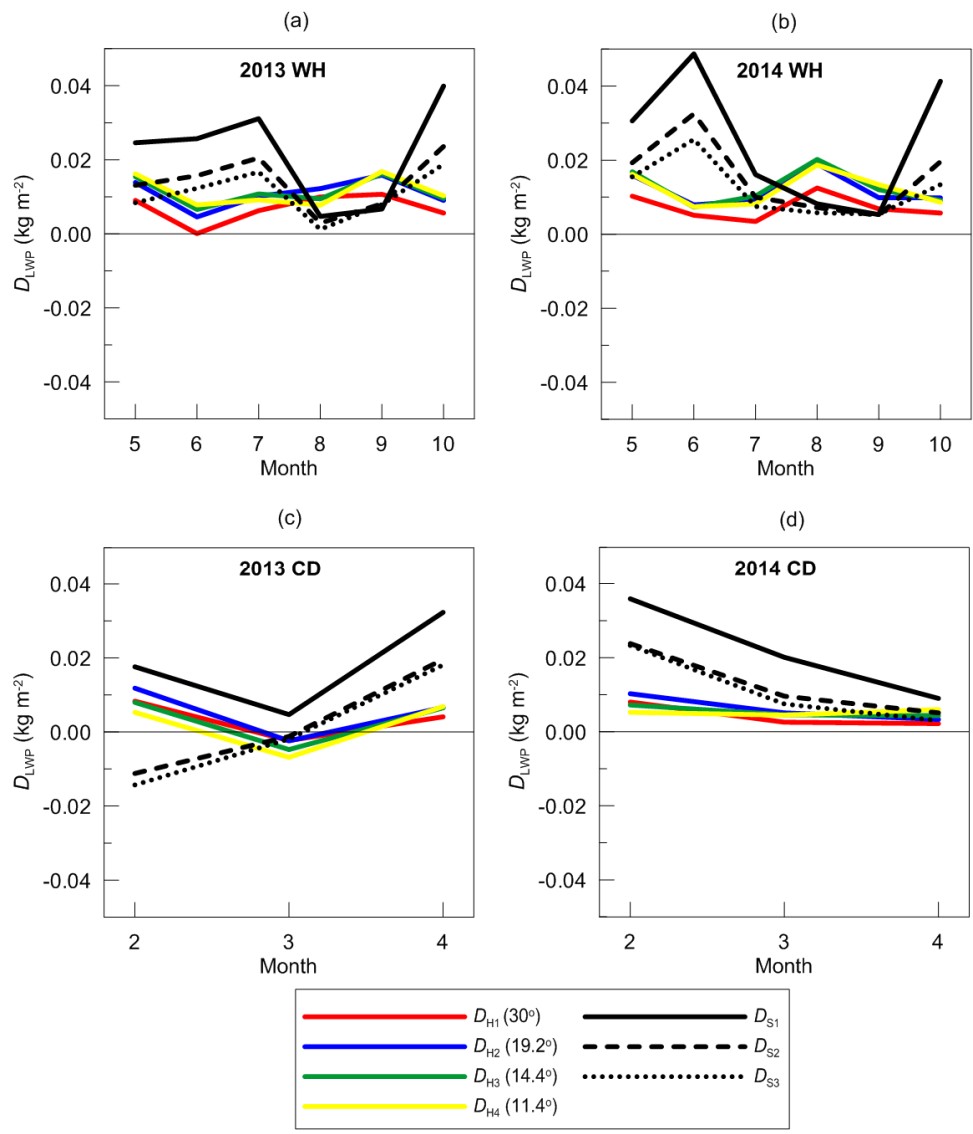

**Figure 12: Monthly mean land-sea LWP difference $D_{LWP}$ as a function of time for various time periods obtained from the satellite and the ground-based observations. $D_{Hj}$ (j=1,…,4) denote $D_{LWP}$ obtained by the HATPRO instrument at four elevation angles (colour lines, see the legend). $D_{Sj}$ (j=1,2,3) denote $D_{LWP}$ obtained by the SEVIRI instrument and calculated by three different formulae, see the text.**

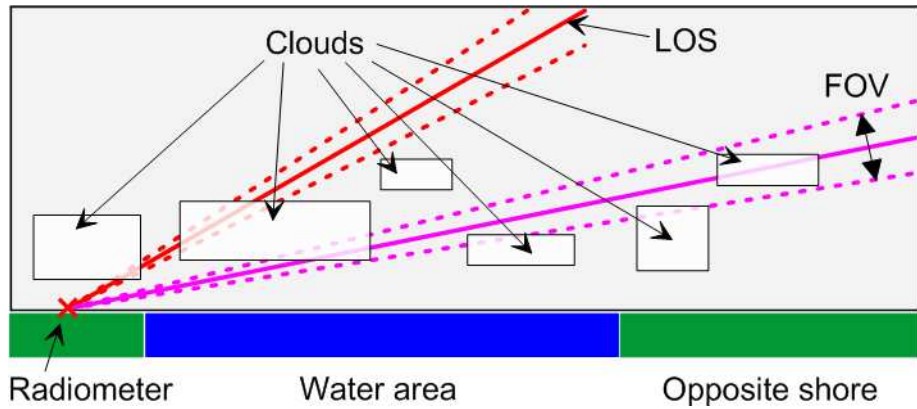

**Fig. A: Possible configurations of the observational geometry in case of scattered clouds (a schematic illustration). Solid lines designate the line-of-sight (LOS) of the observations at various elevation angles. Dashed lines show the field-of-view (FOV) of the radiometer.**

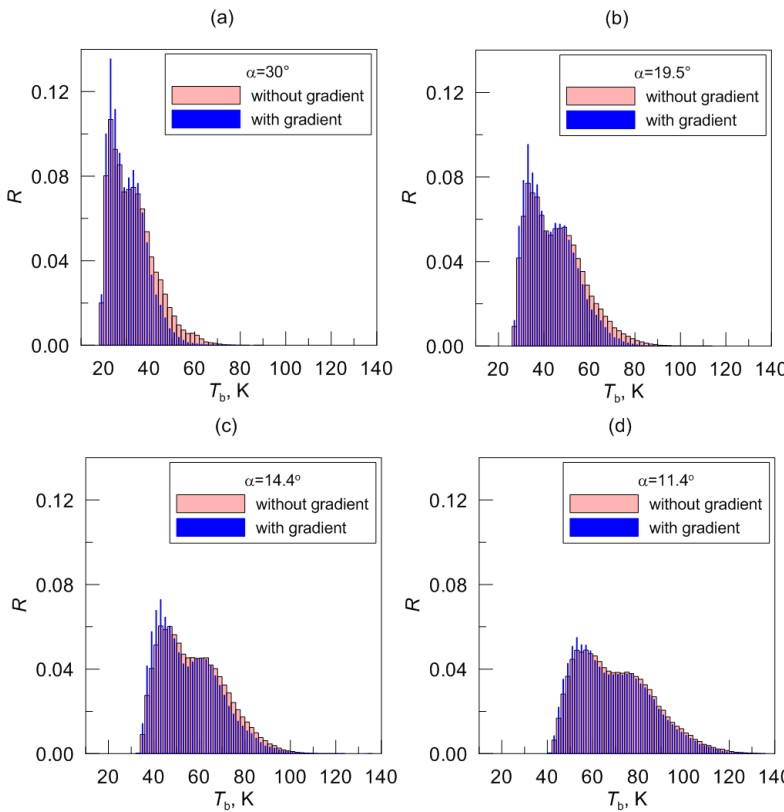

**Fig. B: Statistical distributions (in terms of relative frequency of occurrence R) of brightness temperatures at 31.4 GHz simulated for four elevation angles and for two situations: one with existing LWP land-sea gradient and another without such gradient. Input data: the Monte Carlo model of scattered clouds.**

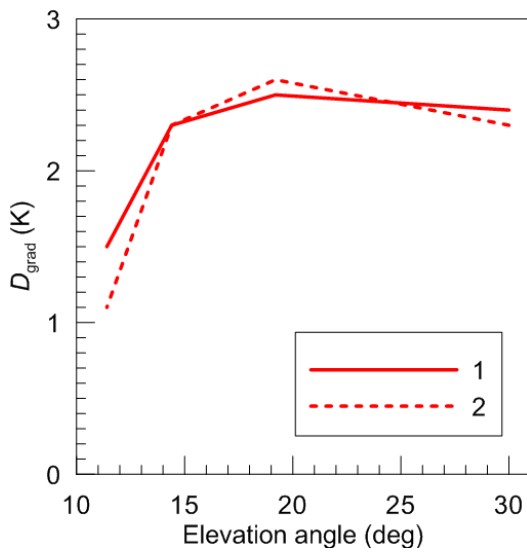

**Fig. C: The LWP gradient signal $D_{grad}$ as a function of the elevation angle at 31.4 GHz. Input data: the Monte Carlo model of scattered clouds. Solid line (1) corresponds to the results obtained with account for FOV; dashed line corresponds to the results obtained when FOV is neglected.**

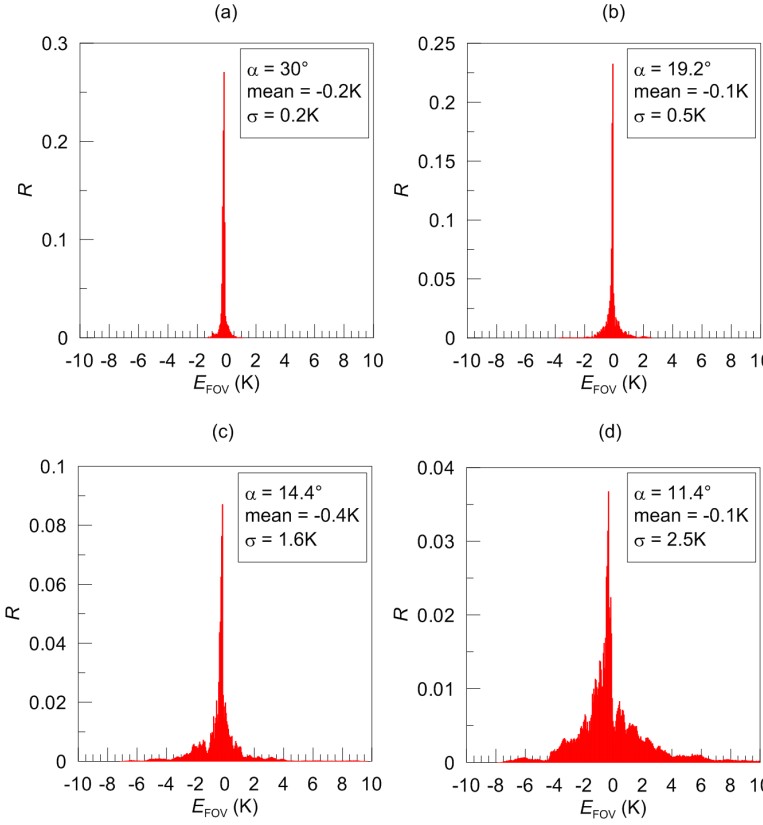

**Fig. D: Statistical distributions (in terms of relative frequency of occurrence *R*) of brightness temperature difference $E_{FOV}$ "$T_B$ neglecting FOV minus $T_B$ accounting for FOV" at 31.4 GHz simulated for four elevation angles. Input data: the Monte Carlo model of scattered clouds.**

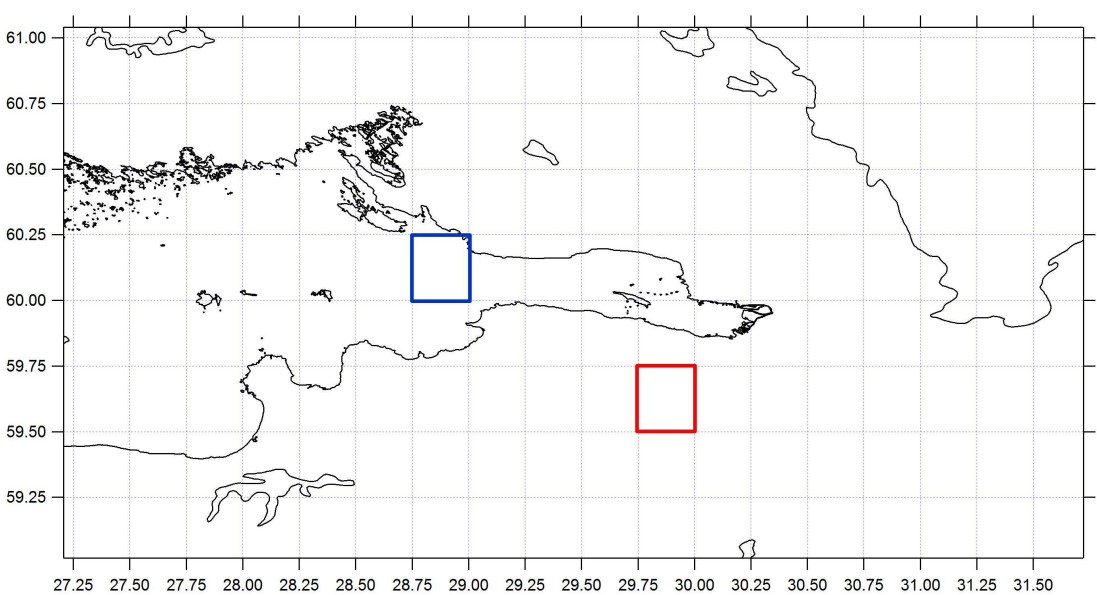

**Fig. E: The map showing the geographical location of the reanalysis data on LWP for the land surface (red) and for the water body (blue).**

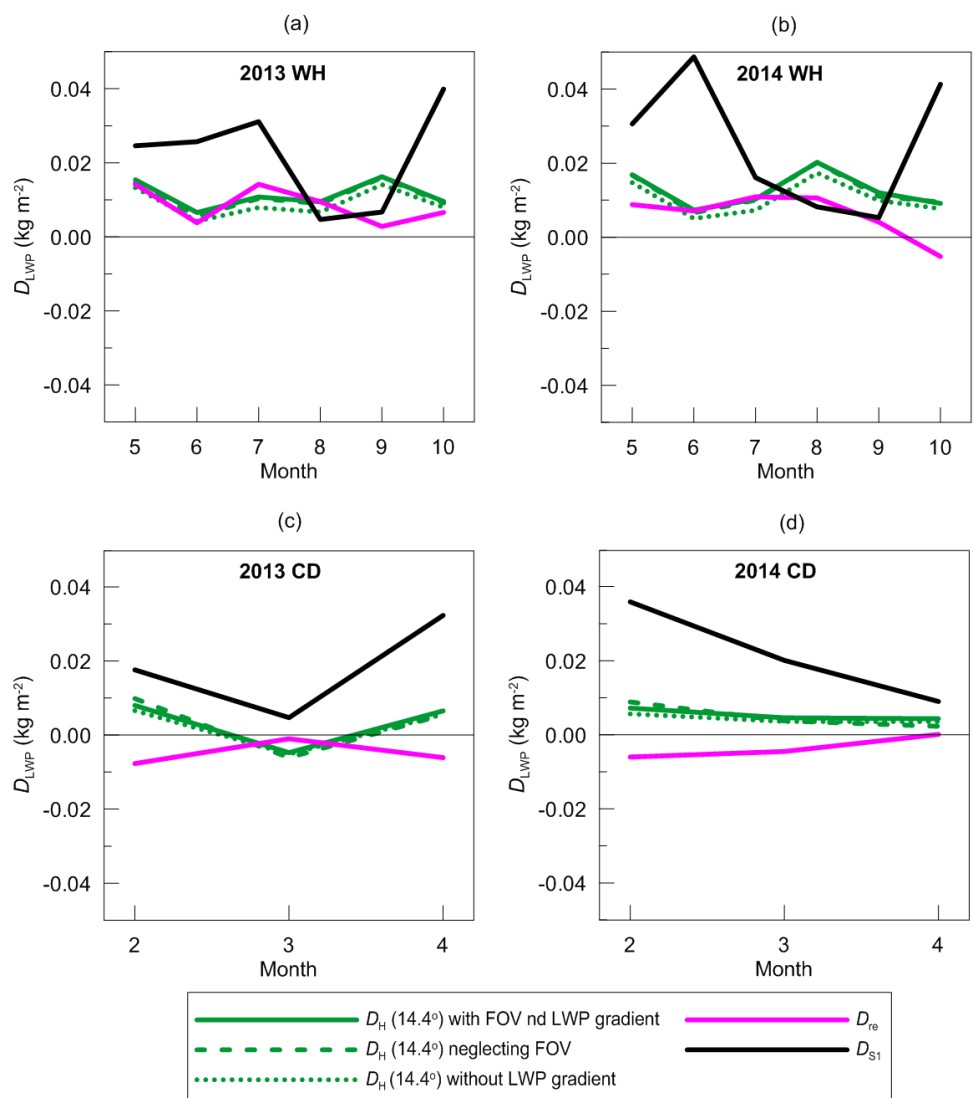

**Figure F: Monthly mean land-sea LWP difference $D_{LWP}$ as a function of time for various time periods obtained from the satellite and the ground-based observations. $D_H$ denotes $D_{LWP}$ obtained by the HATPRO instrument at the elevation angle 14.4° for three scenarios of training the regression algorithm (green lines, see the legend). $D_{S1}$ denotes $D_{LWP}$ obtained by the SEVIRI instrument and calculated by formula (6). $D_{re}$ is the LWP land-sea gradient provided by the ECMWF reanalysis.**