# Peer review of "Detection of the cloud liquid water path horizontal inhomogeneity in a coastline area by means of ground-based microwave observations: feasibility study"

_Atmospheric Measurement Techniques, 2020_

## Referee Comment (RC1) · Anonymous Referee #1 · 12 Mar 2020

In this paper, the authors want to analyze liquid water path (LWP) gradients in a coastal area based on microwave radiometer (MWR) measurements. While the topic in general is of interest, I have substantial concerns about the paper in its present form. The main issues are related to the methodology and conclusions which are drawn.

A large part of the paper is dedicated to the analysis of measured off-zenith brightness temperatures (BTs) in comparison to calculated off-zenith BTs based on the retrieved atmospheric profiles from zenith MWR measurements. The authors state correctly that the BT difference (DTB) which they then derive is related to the gradient in LWP, gra-

dients in T and q as well as further errors and uncertainties. The latter point is really crucial. Large uncertainties are related to the forward calculations they performed using the retrieved T and q profiles (highly smoothed!) and the retrieved LWP. Even if the retrieved LWP is quite accurate, it is still unclear where to place the liquid water vertically. This is not discussed at all and will lead to large uncertainties in the calculated brightness temperatures and brightness temperature differences. This has large implications for the results shown in Figs. 6-10, but the authors merely discuss them. The authors see the problem of disentangling the BT signal of the LWP gradient and that is why the analysis is very qualitative. However, this discussion does not provide a new insight. The conclusions which are drawn could be made without having these measurements: e.g. a liquid cloud located over the instrument with a clear-sky scene around will cause positive DTB values. In my opinion, the whole section on the BT comparison does not provide new insights but rather leaves the reader with many more open questions.

The authors recognize that the best way to proceed is to develop and apply LWP retrieval algorithms and compare LWP directly for the different elevation angles. I agree that this is the way to go, however, again the methodology that they follow to derive the retrieval coefficients is not sound: the authors take the retrieved T and q profiles together with the retrieved LWP again to simulate the BTs for the various elevation angles. Also, here it is not reasonable to use the retrieved profiles for the forward calculations due to the very smoothed T and q profiles (which are thus not representing the realistic atmospheric state). It is again not clear how LWP is vertically distributed. A proper way to generate retrieval coefficients is to use a representative, realistic set of atmospheric profiles from radiosonde or NWP model data.

The MWR measurements/simulations are also set into context to a SEVIRI LWP product. In order to be able to set the results in context to SEVIRI, which views a different scene than HATPRO, a more thorough analysis of the representatively is needed. I am not sure how much can be concluded from the comparison provided in Figs. 11-

12. Yes, on the one hand, SEVIRI and the MWR reveal similar signatures to some extent, on the other hand there are also quite differences. It is totally unclear if this is due to sampling issues, viewing geometry or methodology. Even if uncertainties are discussed I do not see a robust result that can be provided from this comparison.

In the end, the authors state that: "The main conclusion of the study is the following: the approach to detection of the land-sea LWP gradient from microwave measurements by the HATPRO radiometer operating at the observational site of St.Petersburg State University has been successfully tested and the results confirmed the presence of the horizontal land-sea LWP gradient in the vicinity of the radiometer". When looking at Fig. 12, D_LWP for HATPRO reveals various kinds of differences for LWP in zenith and off-zenith directions. These differences are sometimes positive, sometimes negative but there is not a scientific conclusion which can be drawn in my opinion; at least from the results which are presented and considering all the uncertainties which are prevailing in the methodology. Thus, the paper does not provide substantial new insight in this topic in its current form.

In my opinion, the paper needs substantial revision which is beyond major revisions. For this reason, I recommend to decline the manuscript. I suggest to extensively revise the study and encourage the authors to submit a paper at a later stage.

I suggest to concentrate on the analysis of the LWP variability in LWP space and not in BT space and to proper set up multivariate regression-based retrievals for zenith and off-zenith LWP. A physical motivation and discussion for the LWP gradients is currently missing. Why should LWP be enhanced over land than over water? Do you always expect this feature? If the SEVIRI LWP product is used, it needs to be properly introduced and uncertainties discussed (SEVIRI is not the truth!) as well as the representativity of SEVIRI for the HATPRO site and vice versa. Are the LWP pdfs similar for SEVIRI and HATPRO? If case studies with LWP gradients are presented, the physics behind including the role of the meteorological/synoptic situation could shed more light on why certain gradients exist or not. A qualitative analysis is nice but quantifying the LWP

gradient would even add more value to the paper.

---

## Referee Comment (RC2) · Anonymous Referee #2 · 10 Apr 2020

Comments to manuscript AMT-2020-52: Detection of the cloud liquid water path horizontal inhomogeneity in a coastline area by means of ground-based microwave observations: feasibility study by Vladimir S. Kostsov et al.

The paper examines brightness temperatures from a ground-based scanning microwave radiometer to detect the horizontal gradient of LWP between water and land. A few simulations are performed, and the data are interpreted with the help of SEVIRI LWP over land and water. A discussion of possible problems affecting the study outcome is given in the last section of the paper.

General comment

The authors have accomplished a large amount of work on a difficult topic such as the interpretation of off-zenith measurements from a microwave radiometer. Although the concept of using angular measurements to characterize water vapor and liquid water path gradients is feasible, its practical applications are very difficult due to the high variability of the liquid water in the clouds, the inhomogeneity of water vapor, the need to know the cloud location, etc.

In spite of the thorough discussion by the authors, it seems that the only certain result so far is that, under certain very controlled conditions such as those in Fig, 6 and 7, the radiometer contains some qualitative information on the presence of a cloud gradient. However, beyond that, most of the following analysis does not yield any conclusive result. The discussion in section 5 as well does not really provide a definite reason for the figures after Fig. 7.
In addition, the instrument field of view (3 degrees) makes it difficult to interpret the off-zenith measurements if the cloud boundaries are not known. With a 3-degree FOV the radiometer will be sampling a horizontal area of ~ 1 km at 20 km distance when looking up. However, it is not clear if the instrument's field of view was accounted for in the simulations.

I understand that what I am suggesting below is hard because of the effort that was put into this manuscript, however I suggest that the authors rethink the entire methodology used for the analysis and, before they look into the data, they conduct extensive simulations of different scenarios. Detailed suggestions are offered at the end of this review.

Specific comments

Line 196: "The difference between measured and calculated brightness temperatures…"
However, in eq. 1 the difference is between calculated and measured. Please rephrase.

Lines 209-215: This could be a good reason not to use the retrieved profiles as input back to the radiative transfer code to calculate the brightness temperatures off-zenith. Actually, I think the methodology to use the retrieved profiles to re-derive brightness temperatures should be entirely avoided.

Fig. 6-11 If I understand this correctly clouds are not simulated in the calculated brightness temperature. If the cloud base and top are not known, then the brightness temperature information off zenith can only give a very qualitative idea on the presence of clouds.

Fig. 6 and 7 and related discussion. It seems to me that, given the difficulty to interpret the signal below 5 degree, and the fact that it could be related to the interaction between the surface and the atmosphere, it is better to limit the scan to angles > 10 degrees altogether.

Line 302: Fig 7: Should it be Fig. 8?

Fig. 11 and related discussion. I am not sure how useful this Figure is as it is hard to conclude anything from it. The behavior of the two quantities is only weakly correlated, if any.

Fig. 12. As stated by the authors the agreement between satellite and radiometer is not improved by passing from the brightness temperature space to the LWP space. The explanations provided in the next section however are hypothetical and it is hard to really understand what is happening.

I wonder if a better approach for this study would be to use the nearby radiosonde database to simulate a large database of scenarios where clouds with different LWP and different cloud base heights and different geometrical thicknesses are simulated at the radiometer's location and at certain distances from the radiometer. The radiometer field of view needs to be simulated as well. This is especially important for off-zenith measurements.

Brightness temperatures at zenith and different angles for each cloud/distance scenario can then be simulated and a mapping can be established between Tbs differences (zenith – scan) + cloud base height + cloud thickness and PWV, LWP (based on the distance of the simulated cloud) + cloud distance. This could be done with some machine learning given the large number of variables and scenarios and could provide information on whether it is even possible to separate the signal. It would also give an idea of the uncertainty associated with the analysis. The coefficients could then be used with real measurements. Cloud boundaries for the real cases could be derived from satellite or perhaps reanalysis data.

---

## Author Comment (AC1) · 22 May 2020

**The reply to the anonymous referee #1 (RC1)**

We are thankful to the referee for the very detailed analysis of our study. We appreciate the criticism and accept this criticism as very useful for deeper understanding of the combination of a large number of problems relevant to the considered scientific task (the LWP gradient detection by microwave observations). However, we do not agree with several comments and suggestions made by the esteemed referee and below we present our argumentation for that. At the same time we definitely agree with one of the most important statements of the referee relevant to the general conclusion which was made in our study: there was indeed a discrepancy between our too optimistic declaration: "…*the results confirmed the presence of the horizontal land-sea LWP gradient in the vicinity of the radiometer*" and the results which were presented in Fig. 12. When preparing the revised version of our manuscript we chose another scenario (greatly improved) for training of the regression algorithm and we got new results which are now in full agreement with our previous declaration (please see our answers below).

Despite the fact that we argue with several comments and suggestions of the referee, we took all of them into account while preparing the revised version of our manuscript. We agree that we might have described the corresponding issues in the original version of the article not clearly enough. One of the main critical comments of the referee is related to methodology. We hope that the explanations given in the revised version clarify the logic of our research activities and show why we keep the structure of the article and the approach unchanged in the revised version.

Below, the actual comments of the referee are given in **`bold courier font and blue colour`**. The text added to the revised version of the manuscript is marked by red colour.

*Notice: Since both anonymous referees made several similar remarks, our answers to these remarks which are given in both replies are identical.*

**`In this paper, the authors want to analyze liquid water path (LWP) gradients in a coastal area based on microwave radiometer (MWR) measurements. While the topic in general is of interest, I have substantial concerns about the paper in its present form.`**

To the extent of our knowledge the studies devoted to the detection of horizontal inhomogeneities of atmospheric parameters from ground-based passive microwave measurements are not numerous and ours is the first attempt to solve the specific problem relevant to the LWP gradient detection by microwave method in the coastal area. Therefore, we decided that it would be interesting for the scientific community to see the step-by-step analysis of the problem from the very beginning, i.e. starting with the consideration of the forward problem. The task that faces us appeared to be much more complicated than expected when the study had been conceived. We revealed that there are many possible directions of further research both in simulating measurements numerically and in conducting the experiment with modified setup. However, we still have the feeling that the very first results which we obtained will be interesting and useful for the remote sensing scientific community. So, it was the background for our decision to present in the article all our first results along with the identification of problems and possible ways of further development of this research. We do not claim that we obtained the final solution. We demonstrate the complexity of the problem. We partly understand the criticism of the referee towards our paper, but we would like to stress, that the experimental setup of the HATPRO radiometer at our measurements site was initially developed for improving temperature retrievals in the lower layers rather than for solving the problem of the LWP gradient detection. However, we managed to apply these measurements to the task under consideration and got promising results. In order to clarify the logic of our

research, we added the following text at the end of the introduction section in the revised version:

To the extent of our knowledge, the studies devoted to the detection of horizontal inhomogeneities of atmospheric parameters from ground-based passive microwave measurements are not numerous and ours is the first attempt to solve the specific problem relevant to the investigation of the LWP gradient in the coastline area. Therefore, we decided that it would be reasonable to present the step-by-step analysis of the problem starting from the consideration of the forward problem and to demonstrate the complexity of the task that faces us. We used the classical approach to the solution of inverse problem of atmospheric optics: analysis of the forward problem on the basis of simulations, analysis of measured quantities for several test cases, tuning the retrieval algorithm, processing the experimental data with the help of this algorithm, and the comparison of the results to the independent data. Although the concept of using angular measurements to characterize water vapor and liquid water path gradients is feasible, its practical applications are very difficult due to the high variability of the liquid water in the clouds, the inhomogeneity of water vapor, etc.. In addition, we would like to emphasize that the experimental setup of the HATPRO radiometer at our observational site was initially developed for improving temperature retrievals in the lower layers rather than for solving the problem of the LWP gradient detection. However, we managed to apply these measurements to the task under consideration and got promising results.

**A large part of the paper is dedicated to the analysis of measured off-zenith brightness temperatures (BTs) in comparison to calculated off-zenith BTs based on the retrieved atmospheric profiles from zenith MWR measurements. The authors state correctly that the BT difference (DTB) which they then derive is related to the gradient in LWP, gradients in T and q as well as further errors and uncertainties. The latter point is really crucial.**

To our opinion, we can not distinguish any single point as crucial. In the discussion section of the article, we have indicated a large number of other factors which could provide an impact on the considered problem including the sampling scenario, observational geometry, observational condition control, etc..

**Large uncertainties are related to the forward calculations they performed using the retrieved T and q profiles (highly smoothed!) and the retrieved LWP. Even if the retrieved LWP is quite accurate, it is still unclear where to place the liquid water vertically. This is not discussed at all and will lead to large uncertainties in the calculated brightness temperatures and brightness temperature differences. This has large implications for the results shown in Figs. 6-10, but the authors merely discuss them.**

We would like to argue with the esteemed referee against this notion. Indeed, the considered microwave remote sensing method provides highly smoothed T and q profiles and this fact is known and it was quantified in a number of studies with the help of DOFS calculation (Degrees Of Freedom for Signal). This essential nature of the radiative transfer of the downwelling radiation in the considered microwave range exhibits itself both in the forward and inverse problems. The brightness temperature calculations for the zenith and off-zenith geometry are equally insensitive to small scale variations of the parameter distributions along the line of sight. Therefore this smoothing feature does not affect our calculations and relevant conclusions. So, we argue that "**This has large implications for the results shown in Figs. 6-10**". The referee pays attention to the important issue which concerns the placement of the cloud vertically. This issue is closely related to the problem of the profile smoothing and poor spatial resolution of the method. The value of DOFS shows the number of independent pieces of information that can be extracted from microwave observations. For liquid water profile, DOFS is less than 2 that means the small influence of the liquid water distribution on the results of the brightness temperature calculations. This fact indicates implicitly that the placement of the cloud vertically does not play a crucial role in forward calculations and in the solution of the inverse problem. A kind of proof for that is a wide use of regression algorithms for joint IWV (integrated

water vapour) and LWP retrieval from 2-channel observations under the conditions of large uncertainty in the temperature profile and without any information on the cloud placement. However following the comment of the referee that we merely discuss this problem in the article we added the following text in the section "Case study" just before the analysis of Figs. 6-10:

Prior to analysing the cases, we would like to make a note concerning the accuracy of calculations of the brightness temperature difference. These calculations use the temperature, humidity and cloud liquid water profiles retrieved from zenith observations as an input. It is well known that the ground-based microwave method has rather poor spatial resolution which yields smoothed profiles and the very large uncertainty of the vertical placement of a cloud. This fact is known and it was quantified in a number of studies with the help of DOFS calculation (Degrees Of Freedom for Signal which show the number of independent pieces of information that can be extracted from observations). This essential feature of the transfer of the downwelling microwave radiation in the considered spectral region exhibits itself both in the forward and inverse problems. The brightness temperature calculations for the zenith and off-zenith geometry are equally insensitive to small scale variations of the parameter distributions along the line of sight. Therefore this smoothing feature does not affect our calculations and relevant conclusions. The current version of the retrieval setup assumes the placement of a cloud inside the 0.5-5.5 km altitude range (low and medium clouds). Outside this range, the cloud liquid water profile is constrained to zero values. The workability of this retrieval setup has been confirmed in the study devoted to cross-validation of different methods of the LWP retrieval (Kostsov et al., 2018a). For liquid water profile, DOFS is less than 2 that means the small influence of the liquid water distribution on the results of the brightness temperature calculations. This fact indicates implicitly that the placement of the cloud does not play a crucial role in forward calculations and in the solution of the inverse problem. Also, a kind of proof for that is a wide use of regression algorithms for joint IWV (integrated water vapour) and LWP retrieval from 2-channel observations under the conditions of large uncertainty of the temperature profile and without any information on the cloud vertical location. Based on the above mentioned reasons, we consider the applied radiative transfer model accurate enough for making comparisons between measured and calculated brightness temperature values. Also, it is important to note that most of the cases which were selected for analysis are characterized by clear sky conditions over the water area, therefore the cloud placement error is absent for the off-zenith calculations.

As far as the issue of the cloud placement is concerned, we note that this placement (not only vertical, but also horizontal) becomes very important for scattered clouds with horizontal size smaller than the size of the water body under investigation. This is due to the specific off-zenith observational geometry. In the revised version of the article we discuss this circumstance in the new subsection 2.2 on the basis of extensive modelling of scattered clouds and corresponding radiative transfer calculations:

[revised manuscript text omitted]

**The authors see the problem of disentangling the BT signal of the LWP gradient and that is why the analysis is very qualitative. However, this discussion does not provide a new insight. The conclusions which are drawn could be made without having these measurements: e.g. a liquid cloud located over the instrument with a clear-sky scene around will cause positive DTB values. In my opinion, the whole section on the BT comparison does not provide new insights but rather leaves the reader with many more open questions.**

If we understand the referee's opinion correctly, it refers to the section "Case study". We can not agree with this opinion and our reasons are the following:

1) Forward calculations and their comparisons with measurements (analysis in the measurement domain) are very important and in many studies they are a first and an essential step before solving an inverse problem. They are especially useful when considering the multi-parameter inverse problems which physically are formulated as ill-posed. The solution of such problems implies the application of a priori information which can affect the result to a great extent. Besides, in case multiple parameters are retrieved simultaneously, their retrieval errors are coupled in a complex way. These two factors can make the analysis in the domain of sought parameters difficult and ambiguous. Therefore, we start with the analysis in the measurement domain for better understanding of the useful and interfering signals.

2) We guess that the referee refers to our conclusion #1 in the end of section 3 when stating that "**The conclusions which are drawn could be made without having these measurements**". This statement of the esteemed referee does not seem so obvious since: (a) clouds are atmospheric objects, which are characterised by extremely large spatial and temporal variability; (b) probably, the position of the radiometer with respect to the coastline and the experimental setup and geometry are not optimal for the considered task. Therefore, the model simulations should be verified by comparison with experimental data. Besides the theoretical prediction of the value of useful signals should be compared to the experimental data.

3) The referee makes the remark about very qualitative character of our analysis. This is correct to a certain extent since the true state of the atmosphere over the water body (the Neva bay) was unknown: the SEVIRI instrument provides averaged data on LWP, and there was no information on pressure, temperature and humidity profiles. Obviously, quantitative analysis is problematic under such circumstances, but this is not our fault. We managed however to make estimations of the useful and interfering signals.

However, following the referee's comment we added a paragraph in the beginning of section 3:

Forward calculations and their comparisons with measurements are the preliminary and essential steps before solving inverse problems in many studies. Analysis in the measurement domain can be especially useful when considering the multi-parameter inverse problems which physically are ill-posed. The solution of such problems implies the application of a priori information which can affect the result to a great extent. Besides, in case multiple parameters are retrieved simultaneously, their retrieval errors are coupled in a complex way. These two factors can make the analysis in the domain of sought parameters difficult and ambiguous. Therefore we start with the analysis in the measurement domain for better understanding of the useful and interfering signals. Since clouds are atmospheric objects which are characterised by extremely large spatial and temporal variability and since the experimental setup and geometry

were not optimised for considered task, the model simulations should be verified by comparison with experimental data. In addition, the theoretical prediction of the value of useful signal should be compared to the experimental data.

Additionally, we modified the conclusion #1 in section 3:

Concluding this section, we can formulate the following statements:

1) As predicted, the LWP land-sea gradient (higher LWP over land, lower LWP over water) is detectable and shows up as positive values of the difference between modelled and measured brightness temperatures of the MW radiation. These positive values can be seen in the whole considered range of elevation angles (4.8°-30°). The experiment revealed that the magnitude of the useful signal ($D_{grad}$) can vary from 2 K to 24 K depending on elevation angle and LWP land-sea difference (as it is provided by the SEVIRI satellite instrument). Obviously, thorough quantitative analysis is problematic due to the fact that the true state of the atmosphere over the water body (the Gulf of Finland) was unknown: the SEVIRI instrument provided averaged data on LWP, and there was no information on corresponding pressure, temperature, humidity profiles and type of cloudiness.

**The authors recognize that the best way to proceed is to develop and apply LWP retrieval algorithms and compare LWP directly for the different elevation angles. I agree that this is the way to go, however, again the methodology that they follow to derive the retrieval coefficients is not sound: the authors take the retrieved T and q profiles together with the retrieved LWP again to simulate the BTs for the various elevation angles. Also, here it is not reasonable to use the retrieved profiles for the forward calculations due to the very smoothed T and q profiles (which are thus not representing the realistic atmospheric state). It is again not clear how LWP is vertically distributed. A proper way to generate retrieval coefficients is to use a representative, realistic set of atmospheric profiles from radiosonde or NWP model data.**

We do not agree with this statement. Above, we have already presented our opinion about the problem of profile smoothing and the cloud vertical placement in general and about their influence on the results of the forward calculations in particular. As we have already noticed, the more important problem is the cloud horizontal size and placement in case of scattered clouds. In the revised version of the manuscript, we applied our Monte Carlo model of scattered clouds for the derivation of regression coefficients. We used these new regression coefficients and added the retrieval results to the plots which show monthly means of the LWP gradient, please see our answers below.

**The MWR measurements/simulations are also set into context to a SEVIRI LWP product. In order to be able to set the results in context to SEVIRI, which views a different scene than HATPRO, a more thorough analysis of the representatively is needed.**

We strongly disagree with this remark of the referee. First of all, thorough comparison of the HATPRO and SEVIRI data on LWP has already been done in two previous papers by the authors' team published in AMT. The references to these papers are given in the present article in the proper context. We do not think that it is necessary to reproduce already published results. However, addressing this remark of the referee, in the middle of section 4 after the formulae (6, 7, 8) we added a short note in order to emphasize the agreement between the HATPRO and SEVIRI data which had been demonstrated previously:

We would like to emphasize that the extensive and thorough comparison of the HATPRO and SEVIRI data on LWP for pixel 243 has already been made and the results have been published (Kostsov et al., 2018b, 2019). Good agreement for daily mean LWP of the ground-based and satellite data has been revealed. Moreover, the cross-comparison of the HATPRO LWP data with the data from two space-borne instruments SEVIRI and AVHRR confirmed the agreement not only for averaged values, but also for single measurements (Kostsov et al.,

2019). To date, there were no attempts to compare the satellite and ground-based data on LWP over water surfaces. However, the validity of the satellite data over large water bodies was confirmed implicitly by the comparison of the SEVIRI and AVHRR results over the Gulf of Finland and the Lake Ladoga (Kostsov et al., 2019).

**I am not sure how much can be concluded from the comparison provided in Figs. 11-12. Yes, on the one hand, SEVIRI and the MWR reveal similar signatures to some extent, on the other hand there are also quite differences. It is totally unclear if this is due to sampling issues, viewing geometry or methodology. Even if uncertainties are discussed I do not see a robust result that can be provided from this comparison.**

We agree with this remark of the referee. The approach to training the regression algorithm which we had applied previously appeared to be ineffectual (we trained the algorithm separately for each of the considered seasons and years and considered the overcast case only neglecting scattered clouds with varying horizontal and vertical extent). When preparing the revised version of the manuscript, we made thorough forward modelling of scattered clouds and on the basis of this modelling we trained the regression algorithm. The proper training yielded new retrieval results which are robust and clearly show the presence of the LWP land-sea gradient and its seasonal features. We added the comparison with the reanalysis data which showed good agreement between the microwave data and reanalysis data. A large part of section 4 has been changed. The new text and figures are presented here:

[revised manuscript text omitted]
". When looking at Fig.12, D_LWP for HATPRO reveals various kinds of differences for LWP in zenith and offzenith directions. These differences are sometimes positive, sometimes negative but there is not a scientific conclusion which can be drawn in my opinion; at least from the results which are presented and considering all the uncertainties which are prevailing in the methodology. Thus, the paper does not provide substantial new insight in this topic in its current form.**

We agree in general with this remark but we partly disagree with the conclusion of the esteemed referee that "**the paper does not provide substantial new insight in this topic in its current form**". Our study tackles not one single topic of a kind "does the LWP land-sea gradient exist or not?" but a variety of problems and aspects relevant to passive microwave remote sensing using off-zenith geometry. Our study is pioneering in solving the specific task of the LWP land-sea gradient detection and therefore it is natural that some questions are left open. We would like to draw the attention of the esteemed referee to the title of our paper: "feasibility study". Our study is self-consistent in this respect and new insights which are present in the paper refer to the experimental results and their all-round analysis. As far as our conclusion is concerned, we have already mentioned in the beginning of our reply that we definitely agree with one of the most important statements of the referee relevant to the general conclusion which was made in our study: there was indeed a discrepancy between our too optimistic declaration: "*…the results confirmed the presence of the horizontal land-sea LWP gradient in the vicinity of the radiometer*" and the results which were presented in Fig. 12. When preparing the revised version of our manuscript we chose another scenario (greatly improved) for training of the regression algorithm and we got new results (described above) which are now in full agreement with our previous declaration. However we have rearranged the conclusion section accounting for the new retrieval results:

[revised manuscript text omitted]

**In my opinion, the paper needs substantial revision which is beyond major revisions. For this reason, I recommend to decline the manuscript. I suggest to extensively revise the study and encourage the authors to submit a paper at a later stage.**

The referee suggests to extensively revise the study beyond major revision and to resubmit it. First of all we would like to stress that the study is based on the experimental data obtained during several years – brightness temperatures measured from ground at different frequencies. The results relevant to zenith observational mode have been successfully verified, checked, validated and compared with independent data previously. There are no reasons to have doubts in the quality of brightness temperature values obtained in the off-zenith mode. These results can not be revised and, if we understand correctly, the referee means the revision of the data interpretation. We insist that it is a matter of authors choice what approaches and methods to apply for the interpretation and what sequence of activities to select. We have chosen the analysis of the forward problem first and the solution of the inverse problem next. This is a classical sequence. The esteemed referee does not qualify any our approaches and results as "erroneous". The critical remarks refer mainly to the fact that the results are not convincing enough. We do not think that it is the serious reason for declining the paper, especially taking into account the fact that our work is pioneering for the specific task of LWP land-sea gradient detecting by ground-based passive microwave radiometry. As we have shown in the section 5, there are a large number of various factors which affect the results. In the revised version of our manuscript we present the retrieval results which were obtained by the newly trained regression

algorithm. We got promising results. We think that there are still many possibilities for further improvements, but nevertheless the existing results are interesting and useful.

**I suggest to concentrate on the analysis of the LWP variability in LWP space and not in BT space and to proper set up multivariate regression-based retrievals for zenith and off-zenith LWP. A physical motivation and discussion for the LWP gradients is currently missing. Why should LWP be enhanced over land than over water? Do you always expect this feature? If the SEVIRI LWP product is used, it needs to be properly introduced and uncertainties discussed (SEVIRI is not the truth!) as well as the representativity of SEVIRI for the HATPRO site and vice versa. Are the LWP pdfs similar for SEVIRI and HATPRO? If case studies with LWP gradients are presented, the physics behind including the role of the meteorological/synoptic situation could shed more light on why certain gradients exist or not.**

In this comment, the referee puts out many suggestions for the revision of the paper. Several of these issues have already been mentioned in the original version of the manuscript and discussed briefly. The extent analysis of the topics suggested by the referee is beyond the scope of the present study which does not have the goal to embrace and analyse in detail the whole variety of primary and secondary problems which arise (relevant both to the experimental part and to the interpretation part). Our short comments to the referee's suggestions are the following:

1) We insist that it is the matter of the authors' choice to make analysis booth in the BT space and LWP space.
2) Now, the regression algorithm trained in different way has been applied and the results are included in the revised version.
3) Physical motivation for LWP gradient is given in the Introduction with proper references to literature.
4) Extended analysis of the LWP gradient as observed from satellites is beyond the scope of our study.
5) The quality of the SEVIRI data and the comparisons of the ground-based and space-borne data are discussed in the revised version (see our respective answer above).
6) We agree that in the case studies the analysis of synoptic situations would be interesting but it is not so important as the cloud size and horizontal and vertical location which is studied in the revised version.

**A qualitative analysis is nice but quantifying the LWP version gradient would even add more value to the paper.**

We are thankful to the referee for the high estimate given to the section 5 (we guess that this comment refers to this section).

Concluding our reply we would like to thank the referee once again for the comments which indeed helped to improve our manuscript. We edited the acknowledgement section accordingly:

The authors are grateful to two anonymous referees for making very insightful remarks and for introducing several useful ideas which helped greatly to improve the manuscript.

Vladimir Kostsov
on behalf of all co-authors

**Note: For convenience, in the revised version of the manuscript the new figures have their own numeration (by letters) and are placed at the end of the manuscript.**

---

## Author Comment (AC2) · 22 May 2020

**The reply to the anonymous referee #2 (RC2)**

We are grateful to the referee for the very attentive reading of our manuscript and for many insightful remarks. We accept part of the criticism, but argue with several comments and general conclusion made by the referee. While preparing the revised version of our article, we took into account all comments made by the referee.

Below, the actual comments of the referee are given in **`bold courier font and blue colour`**. The text added to the revised version of the manuscript is marked by red colour.

*Notice: Since both anonymous referees made several similar remarks, our answers to these remarks which are given in both replies are identical.*

**`Attached are my comments to the manuscript amt-2020-52. After careful reading multiple times it is my opinion that the methodology used in the paper is not adequate to provide a sound interpretation of the data. Because of the complexity of the topic I suggest that the authors rethink the way they have approached the problem, perhaps doing more simulations. I provide more details in the attached comments and offer some suggestions as well, hoping that they can be useful.`**

The esteemed reviewer makes general conclusion about the inadequacy of the methodology which we used in our study. We can not agree with this conclusion. We used the classical approach to the solution of inverse problem of atmospheric optics: analysis of the forward problem on the basis of simulations, analysis of measurements in several test cases, tuning the retrieval algorithm, processing the experimental data with the help of this algorithm, and the comparison of the results to the independent data. We obtained consistent results. The fact that a number of questions still remain open does not mean that the interpretation had not been sound. Contrariwise, it indicates the complexity of the problem and shows the ways for further research. The referee advises to rethink the way of approaching the problem. We would like to stress that our study is based on experimental multi-year data. Though the experimental setup of the HATPRO radiometer at our measurements site was initially developed for improving temperature retrievals in the lower layers rather than for solving the problem of the LWP gradient detection and so it was not optimal, nevertheless we managed to apply these measurements to the task under consideration and got some promising results. We have already shown in the discussion section that the experimental setup (geometry, sampling, etc.) may have a large impact on the obtained results. Therefore, "rethinking of the approach" may imply also the transfer to a new measurement scenario. This can be done, of course, but we think that it is beyond the scope of the present study. The current study, to our opinion, is complete, non-contradictory and contains new results. To the extent of our knowledge the studies devoted to the detection of horizontal inhomogeneities of atmospheric parameters from ground-based passive microwave measurements are not numerous and ours is the first attempt to solve the specific problem relevant to the LWP gradient in the coastal area. In order to clarify the motivation for our study and the applied methodology, we added the following text at the end of the introduction section:

> To the extent of our knowledge, the studies devoted to the detection of horizontal inhomogeneities of atmospheric parameters from ground-based passive microwave measurements are not numerous and ours is the first attempt to solve the specific problem relevant to the investigation of the LWP gradient in the coastline area. Therefore, we decided that it would be reasonable to present the step-by-step analysis of the problem starting from the consideration of the forward problem and to demonstrate the complexity of the task that faces us. We used the classical approach to the solution of inverse problem of atmospheric optics: analysis of the forward problem on the basis of simulations, analysis of measured quantities for several test cases, tuning the retrieval algorithm, processing the experimental data with the help

of this algorithm, and the comparison of the results to the independent data. Although the concept of using angular measurements to characterize water vapor and liquid water path gradients is feasible, its practical applications are very difficult due to the high variability of the liquid water in the clouds, the inhomogeneity of water vapor, etc.. In addition, we would like to emphasize that the experimental setup of the HATPRO radiometer at our observational site was initially developed for improving temperature retrievals in the lower layers rather than for solving the problem of the LWP gradient detection. However, we managed to apply these measurements to the task under consideration and got promising results.

**General comment**

**The authors have accomplished a large amount of work on a difficult topic such as the interpretation of off-zenith measurements from a microwave radiometer. Although the concept of using angular measurements to characterize water vapor and liquid water path gradients is feasible, its practical applications are very difficult due to the high variability of the liquid water in the clouds, the inhomogeneity of water vapor, the need to know the cloud location, etc.**

We completely agree with this remark of the referee. The task that faces us appeared to be much more complicated than expected when the study had been conceived. We revealed that there are many possible directions of further research both in simulating measurements numerically and in conducting the experiment with modified setup.

**In spite of the thorough discussion by the authors, it seems that the only certain result so far is that, under certain very controlled conditions such as those in Fig, 6 and 7, the radiometer contains some qualitative information on the presence of a cloud gradient. However, beyond that, most of the following analysis does not yield any conclusive result. The discussion in section 5 as well does not really provide a definite reason for the figures after Fig. 7.**

First of all, we dare to suspect that the referee meant not Figs. 6 and 7, but some others. Figs. 6 and 7 correspond to clear sky conditions everywhere and the cloud gradient can not be expected there. We strongly disagree with the statement made by the esteemed referee that "**most of the following analysis does not yield any conclusive result**". To the best of our knowledge, our results are the first ones which directly refer to the practice of solving the specific problem of LWP gradient detection in the coastline area by ground-based MW method. The outcome of the research is unknown. Also, we would like to stress that our research is based on the experimental data. In this respect any obtained estimations are conclusive since they provide values and data which were unknown before. We can give some examples of the results which we consider conclusive: (a) estimations of the magnitude of the useful signal; (b) the results of $T_b$ measurements in special selected cases; (c) the estimations of the LWP gradient effect and the analysis of error components. However we admit the fact that to some extent it is a philosophical question: what result can be considered conclusive and what result can not…

**In addition, the instrument field of view (3 degrees) makes it difficult to interpret the off-zenith measurements if the cloud boundaries are not known. With a 3-degree FOV the radiometer will be sampling a horizontal area of ~ 1 km at 20 km distance when looking up. However, it is not clear if the instrument's field of view was accounted for in the simulations.**

We definitely agree that we should have addressed this issue in our manuscript. We did not take the FOV of the radiometer into account. In the revised version we performed extensive simulations of measurements accounting for FOV and demonstrate the validity of our previous results. Please, see below our answer to the remark which concerns extensive simulations.

*I understand that what I am suggesting below is hard because of the effort that was put into this manuscript, however I suggest that the authors rethink the entire methodology used for the analysis and, before they look into the data, they conduct extensive simulations of different scenarios. Detailed suggestions are offered at the end of this review.*

In the beginning of our reply we have already argued with the referee on the point of "rethinking the entire methodology". We can not understand the criticism expressed by the referee towards our methodology. The esteemed referee does not qualify any our approaches and results as "erroneous". We have the feeling that the referee expects that minor improvements in setting up the forward and inverse calculations will lead to definite answers which will change the results dramatically. Our opinion is opposite. However, as far as extensive simulations of different scenarios are concerned, we thank the referee for this suggestion, we consider this suggestion as very useful which can improve the estimations made in course of the analysis of the forward problem. We took this suggestion into account in the revised version. We performed extensive modelling of scattered clouds and made corresponding radiative transfer calculations. The new subsection 2.2 was added to the manuscript:

[revised manuscript text omitted]

**Specific comments**

**Line 196: "The difference between measured and calculated brightness temperatures…" However, in eq. 1 the difference is between calculated and measured. Please rephrase.**

Corrected.

**Lines 209-215: This could be a good reason not to use the retrieved profiles as input back to the radiative transfer code to calculate the brightness temperatures off-zenith. Actually, I think the methodology to use the retrieved profiles to re-derive brightness temperatures should be entirely avoided.**

We would like to argue on that point. It is noticed in lines 209-215: "*Here, one important note should be made: the retrieval errors for profiles have random and systematic components (the latter is caused mainly by a priori information used for retrievals). As a result, the term $D_{err}$ might consist of both components also.*" So, it is

not necessarily that the impact of the corresponding error will be large if averaged quantities are analysed when the random error component is strongly suppressed.

**Fig. 6-11 If I understand this correctly clouds are not simulated in the calculated brightness temperature. If the cloud base and top are not known, then the brightness temperature information off zenith can only give a very qualitative idea on the presence of clouds.**

First of all, it is important keep in mind that in these calculations there are no atmospheric parameters which are simulated. We just take the parameters retrieved from zenith observations, assume that they are the same over water body and calculate $T_b$s for off-zenith geometries. These $T_b$s are then compared to measured $T_b$s. Second, all specially selected cases refer to situations with clear sky over the water body. It does not really matter where a cloud is placed vertically when we simulate off-zenith observations. The useful signal is detectable and we show this.

**Fig. 6 and 7 and related discussion. It seems to me that, given the difficulty to interpret the signal below 5 degree, and the fact that it could be related to the interaction between the surface and the atmosphere, it is better to limit the scan to angles > 10 degrees altogether.**

We completely agree with this advice of the referee. In the study of seasonal features and when making the retrievals we use the limit for the elevation angles 10 degrees.

**Line 302: Fig 7: Should it be Fig. 8?**

Yes, corrected.

**Fig. 11 and related discussion. I am not sure how useful this Figure is as it is hard to conclude anything from it. The behavior of the two quantities is only weakly correlated, if any.**

To our opinion, there are similar patterns in temporal behaviour of the compared quantities and these similarities are important. The retrieval results (LWP gradient values) which are presented in the revised version of the article exhibit similarities for the cold season but not for the warm season while the quantities in Fig. 7 demonstrate similar features just during the warm season.

**Fig. 12. As stated by the authors the agreement between satellite and radiometer is not improved by passing from the brightness temperature space to the LWP space. The explanations provided in the next section however are hypothetical and it is hard to really understand what is happening.**

We agree with this remark of the referee. The approach to training the regression algorithm which we had applied previously appeared to be ineffectual (we trained the algorithm separately for each of the considered seasons and years and considered the overcast case only neglecting scattered clouds with varying horizontal and vertical extent). When preparing the revised version of the manuscript, we made thorough forward modeling of scattered clouds (as suggested by the referee) and on the basis of this modeling we trained the regression algorithm. The proper training yielded new retrieval results which are robust and clearly show the presence of the LWP land-sea gradient and its seasonal features. We added the comparison with the reanalysis data which showed good agreement between the microwave data and reanalysis data. A large part of section 4 has been changed. The new text and figures are presented here:

[revised manuscript text omitted]

I wonder if a better approach for this study would be to use the nearby radiosonde database to simulate a large database of scenarios where clouds with different LWP and different cloud base heights and different geometrical thicknesses are simulated at the radiometer's location and at certain distances from the radiometer. The radiometer field of view needs to be simulated as well. This is especially important for off-zenith measurements.

Similar remark has already been made by the referee. We agree with this remark and we are grateful to the referee for the hint to simulate large database. We did it accounting for FOV and simulated scattered clouds (see our answer above). However we used the atmospheric parameters from the HATPRO retrievals rather than from radiosondes. Since the vertical resolution of ground-based microwave remote sensing is poor, we do not see the necessity to use radiosonde profiles.

Brightness temperatures at zenith and different angles for each cloud/distance scenario can then be simulated and a mapping can be established between Tbs differences (zenith – scan) + cloud base height + cloud thickness and PWV, LWP (based on the distance of the simulated cloud) + cloud distance. This could be done with some machine learning given the large

**number of variables and scenarios and could provide information on whether it is even possible to separate the signal. It would also give an idea of the uncertainty associated with the analysis. The coefficients could then be used with real measurements. Cloud boundaries for the real cases could be derived from satellite or perhaps reanalysis data.**

We are grateful to the referee for this hint. Partly, we have implemented it while preparing the revised version of our manuscript and got new robust results. However we would like to note that implementing of all suggestions of the esteemed referee seems to turn into a separate study (may be not only one study, but several). Our study is pioneering in solving the specific task of the LWP land-sea gradient detection and therefore it is natural that some questions are left open.

Concluding our reply we would like to thank the referee once again for the comments which indeed helped to improve our manuscript. We edited the acknowledgement section accordingly:

> The authors are grateful to two anonymous referees for making very insightful remarks and for introducing several useful ideas which helped greatly to improve the manuscript.

Vladimir Kostsov
on behalf of all co-authors

**Note: For convenience, in the revised version of the manuscript the new figures have their own numeration (by letters) and are placed at the end of the manuscript.**

---

## Author Response (AR2)

**The reply to the anonymous referee #2 (Report#1)**

We are thankful to the referee for the positive assessment of the changes in the revised version of our manuscript. We agree with all remarks and suggestions made by the referee. We took all comments into account while preparing the new revised version of our manuscript.

Below, the actual comments of the referee are given in **`bold courier font and blue colour`**. The text added to the revised version of the manuscript is marked by red colour.

**`The changes made by the authors have improved the manuscript and provided more information to better understand the results. The downside of the additions is that the manuscript is now extremely long.`**

We agree that the manuscript became very long and we tried to reduce it considerably while preparing the new revised version.

**`In the interest of readability, I am giving below some suggestions on possible reductions in a way that hopefully is not detrimental to the understanding of the paper. The suggestions are of course optional for the Authors.`**

We are grateful to the esteemed referee for these suggestions. We tried to follow them.

**`The abstract is very long, can it be reduced? Are all the setup details described in lines 14-22 necessary in the abstract? Similar considerations hold for lines 29-32.`**

We agree with this suggestion. We removed lines 16-22 and 29-32.

**`Introduction: There are 5 pages of introduction. I do understand the need to put the study in perspective, but 5 pages seem a little excessive. The authors should consider shrinking.`**

We agree with this comment. We managed to make the introduction almost two times shorter by removing the description of scanning radiometers and of the methodologies and approaches to detect horizontal inhomogeneities. However, we kept all references relevant to these topics.

**`I am not sure that the entire section 5 adds anything to the analysis or the understanding of the results, as it consists of mostly general considerations that have already been discussed. Sections 5.1, 5.2 and 5.3 can probably be entirely removed and only section 5.4 and 5.5 left and perhaps put in an appendix.`**
**`1. A short justification for the sampling interval can be given in section 2.1 (lines 165 and following);`**
**`2. The orientation of the instrument and effect of the Gulf Finland and lake have already been extensively discussed;`**
**`3. The data processing algorithm has already been discussed in section 4. A quick reference to a physical retrieval can be made in that section to highlight the fact that a better algorithm could be used, although I don't think is going to make a difference on a statistical basis.`**

We are thankful to the referee for the hint to put section 5 in an appendix. We removed the subsection devoted to the orientation of the instrument completely, but we decided to keep other subsections in order to demonstrate all major problems together. However, we strongly truncated subsections devoted to sampling interval and processing algorithm following the advice of the referee. As a result, the Appendix occupies much less space than section 5 in the previous version of the manuscript.

So, the new revised version of the manuscript consists of 22 pages of the main text instead of 28 pages of the previous version. The Appendix contains the truncated section 5 and occupies only 3 pages of text. We hope that the readability of the article has been improved without any detriment to understanding.

Vladimir Kostsov
on behalf of all co-authors

**The reply to the anonymous referee #1 (Report#2)**

We are thankful to the referee for the appreciation of our efforts to improve the manuscript. We agree with all remarks and suggestions made by the referee. We took all comments into account while preparing the new revised version of our manuscript.

Below, the actual comments of the referee are given in **`bold courier font and blue colour`**. The text added to the revised version of the manuscript is marked by red colour.

**`I appreciate that the authors revised the manuscript in many ways. Also, the conclusions are now more carefully formulated and uncertainties discussed in detail. I also like that the authors demonstrate how these observations could be used for model evaluation.`**

**`Still, I have one major point concerning the forward simulated brightness temperatures (TBs). In ll. 372-392, uncertainties related to the forward calculations are qualitatively discussed but no quantative measures provided. While for off-zenith calculations, it is difficult to evaluate the uncertainty due to the spatial variability of atmospheric variables, for zenith observations, the calculated TBs can and should be compared to the measured values. If the zenith TB measurements are well reproduced, differences between the calculated and measured off-zenith TBs are mainly related to the spatial inhomogeneity of LWP, water vapor and temperature. However, if the uncertainties in the zenith calculations are bigger than the TB differences caused by the the spatial variability of LWP, q, … (shown in the cases studies in Figs. 6-10), it is difficult to draw any conclusions. I think that it is important to show how accurate the simulated zenith TBs under clear and cloudy conditions are and to provide some numbers here. These numbers need to be set into context to the numbers shown in Figs. 6-10 to gain confidence in the results.`**

We agree with this comment. We added the following text just after the paragraph which is mentioned by the referee (lines 372-392):

In order to quantify the accuracy of our forward calculations, we present the values of the residual between measured brightness temperatures and the brightness temperatures which are calculated using the retrieved profiles of atmospheric parameters for zenith observations. The RMS residual $R_{RMS}$ and the mean residual $R_{mean}$ are calculated for every retrieval separately for seven "humidity channels", for seven "temperature channels", and for all 14 spectral channels of the radiometer. These quantities are used for the data quality control during the routine observations: the results are filtered out if $R_{RMS}$ for all 14 channels is larger than 1 K. The large statistics comprising clear and cloudy conditions and all seasons shows that $R_{RMS}$ and $R_{mean}$ for "humidity channels" which are of primary interest in the present study constitute in average 0.2 K and 0.05 K respectively. So, the $T_B$ measurements are well reproduced. In order to gain confidence in the results relevant to the LWP inhomogeneity, we supposed that it would be reasonable to take the absolute value of the threshold for the "useful signal" in $D_{TB}$ equal to 1 K which is five times larger than the typical $R_{RMS}$ value for "humidity channels". The $D_{TB}$ values exceeding this threshold are mainly related to the horizontal inhomogeneity of atmospheric parameters. We took into account this threshold value when we plotted Figs. 6-10.

**`In general, the manuscript is very long (28 pages text, 18 figures!). I think that some discussions should be shortened but I would ask the authors to check where there is potential to reduce the length of the manuscript.`**

We absolutely agree with this comment. We tried to reduce the article size:

1) In the abstract we removed lines 16-22 and 29-32.

2) We managed to make the introduction almost two times shorter by removing the description of scanning radiometers and of the methodologies and approaches to detecting horizontal inhomogeneities. However we kept all references relevant to these topics.

3) We put section 5 in the appendix. We removed the subsection devoted to the orientation of the instrument completely, but we decided to keep other subsections in order to demonstrate all major problems together. However, we strongly truncated subsections devoted to sampling interval and processing algorithm. As a result, the Appendix occupies much less space than section 5 in the previous version of the manuscript.

So, the new revised version of the manuscript consists of 22 pages of the main text instead of 28 pages of the previous version. The Appendix contains the truncated section 5 and occupies only 3 pages of text. We hope that the readability of the article has been improved without any detriment to understanding.

**In addition, I have a few minor comments:**

**Figs. 6-10: Could you plot as well the LWP value which has been used to calculate the TB values? The SEVIRI LWP values at the HATPRO location is shown but this is not the value used in forward calculations. It think this information would be helpful as well.**

In general, we agree with this comment. When we made Figs. 6-10, we had the same idea: to plot also the LWP values obtained by HATPRO from zenith observations. However finally, after some hesitations, we gave up this idea in order to keep consistency in the data which are used. The reason for that is the following. Plotting the LWP values derived from HATPRO and SEVIRI measurements in the same figure implies their comparison. Multiple studies have shown that proper comparison of the ground-based microwave results with the satellite results require temporal averaging of the ground-based data. In our study we use the HATPRO data collected in the angular scanning mode only. The data are sampled with a 20 minute interval. No averaging is possible for this measurement scenario. And we have the opinion that plotting the instantaneous LWP obtained by HATPRO together with the SEVIRI data would be misleading for the readers. There was another possibility: to use the routine zenith observations with 10 second sampling for the LWP retrieval plus subsequent averaging and to plot these results. However in this case there would be inconsistency in using the HATPRO data obtained in different measurement modes.

We agree with the esteemed referee that the information on LWP derived from HATPRO observations would be helpful, but due to above mentioned reasons we decided to keep Figs. 6-10 unchanged. Besides, modifications of Figs. 6-10 would require additional explanations, which would increase the article size. At the same time there is a strong recommendation of the referee to reduce the length of the manuscript.

**l 81: at "a" synoptic station**

Corrected.

**l 82: The authors of "the" mentioned study**

Corrected.

**l 468: should have produced "a" signal**

Corrected.

**l 699: IPT is the specific retrieval Loehnert et al. (2008) developed, not a general name**

We agree with this comment. We removed mentioning of IPT keeping only the references:

[revised manuscript text omitted]